# Layer-specific morphological and molecular differences in neocortical astrocytes and their dependence on neuronal layers

Darin Lanjakornsiripan[1], Baek-Jun Pior[1], Daichi Kawaguchi[1], Shohei Furutachi[1,2], Tomoaki Tahara[1], Yu Katsuyama[3], Yutaka Suzuki[4], Yugo Fukazawa[5] & Yukiko Gotoh[1]

Non-pial neocortical astrocytes have historically been thought to comprise largely a non-diverse population of protoplasmic astrocytes. Here we show that astrocytes of the mouse somatosensory cortex manifest layer-specific morphological and molecular differences. Two- and three-dimensional observations revealed that astrocytes in the different layers possess distinct morphologies as reflected by differences in cell orientation, territorial volume, and arborization. The extent of ensheathment of synaptic clefts by astrocytes in layer II/III was greater than that by those in layer VI. Moreover, differences in gene expression were observed between upper-layer and deep-layer astrocytes. Importantly, layer-specific differences in astrocyte properties were abrogated in *reeler* and *Dab1* conditional knockout mice, in which neuronal layers are disturbed, suggesting that neuronal layers are a prerequisite for the observed morphological and molecular differences of neocortical astrocytes. This study thus demonstrates the existence of layer-specific interactions between neurons and astrocytes, which may underlie their layer-specific functions.

[1] Graduate School of Pharmaceutical Sciences, The University of Tokyo, Tokyo 113-0033, Japan. [2] Sainsbury Wellcome Centre for Neural Circuits and Behaviour, University College London, London W1T 4JG, UK. [3] Department of Anatomy, Shiga University of Medical Science, Otsu 520-2192, Japan. [4] Department of Computational Biology, Graduate School of Frontier Sciences, The University of Tokyo, Tokyo 277-8561, Japan. [5] Graduate School of Medical Sciences, University of Fukui, Fukui 910-1193, Japan. These authors contributed equally: Darin Lanjakornsiripan, Baek-Jun Pior, Daichi Kawaguchi. Correspondence and requests for materials should be addressed to S.F. (email: s.furutachi@ucl.ac.uk) or to Y.G. (email: ygotoh@mol.f.u-tokyo.ac.jp)

A strocytes are abundant throughout the mammalian central nervous system (CNS) and provide physical and nutritional support to neurons. Recent studies have also revealed that these cells contribute to information processing through regulation of synapse formation and elimination, as well as of synaptic transmission[1–4]. Astrocytes manifest diverse morphological features[5], with those in different CNS regions or progenitor domains having recently been shown to have distinct properties[6–9].

Even within the same progenitor domains, astrocyte diversification may increase the complexity of the corresponding neuronal networks[10]. For example, astrocytes in layer I of the mammalian neocortex are often referred to as pial astrocytes or marginal glia and appear to possess distinct properties with regard to spontaneous $Ca^{2+}$ signaling activity and molecular expression patterns compared with those in the other cortical layers, which are collectively referred to as protoplasmic astrocytes[9,11–13].

Besides astrocyte types that have evolved specifically in higher-order primates[14], protoplasmic astrocytes in layers II–VI of the mammalian neocortex have historically been thought to comprise a homogeneous population. Although, recent studies have indicated the existence of molecularly distinct subpopulations of neocortical protoplasmic astrocytes[15–17], it has remained unclear whether such subpopulations are randomly distributed or associated with specific layers. Given that layer-specific neuronal subtypes play essential roles in cortical circuitry, astrocytes might also be expected to support and modify this circuitry in a layer-specific manner.

We have now performed a comprehensive characterization of the three-dimensional (3D) morphologies and molecular expression patterns of astrocytes in all layers of the mouse somatosensory cortex and found that neocortical astrocytes display morphological and molecular differences corresponding to the laminar organization of neocortical neurons. We also examined how such layer-specific properties of astrocytes might be established during development. Given that neuronal layers are established before astrocyte differentiation, we investigated the role of these layers in the development of layer-specific properties of astrocytes. We thus studied two animal models, *reeler* and *Dab1* conditional knockout (cKO) mice in which neuronal layers are disturbed. In the latter model, *Dab1* was deleted specifically in neurons in order to address whether neuronal layers are responsible for astrocyte diversification. The results suggested a prerequisite role of neuronal layers in establishing the astrocyte diversity in the neocortex.

## Results

**Morphological differences among neocortical astrocytes**. To investigate the heterogeneity of neocortical astrocytes, we first examined the 3D morphology of astrocytes in layers II/III and VI of the mouse primary somatosensory cortex at postnatal day (P) 60. We studied Glast-EMTB-GFP transgenic mice, which express Enconsin microtubule-binding domain-tagged green fluorescent protein (GFP) driven by the *Glast* (*Slc1a3*) promoter and in which microtubules of astrocytes are labeled with GFP throughout all cortical layers, and we applied the CUBIC technique to render brain slices transparent[18,19]. 3D observation revealed that astrocytes in layer II/III differ from those in layer VI in terms of cell orientation and process arborization (Fig. 1a). Layer II/III astrocytes tended to elongate radially (vertically relative to the pial surface), whereas layer VI astrocytes tended to elongate tangentially (horizontally relative to the pial surface). Moreover, the extent of process arborization in layer II/III astrocytes tended to be greater than that in layer VI astrocytes.

We next quantified such morphological differences among astrocytes in all cortical layers including layer I in which astrocytes were previously shown to differ from those in other layers in terms of $Ca^{2+}$ signaling activity and gene expression[11,13], but their morphological differences have remained uncharacterized. To define the cell boundary and visualize the morphology of individual astrocytes as a whole, we therefore studied Glast-EMTB-GFP;Glast-CreER^T2;Rosa-CAG-loxP-stop-loxP(LSL)-tdTomato mice in which tamoxifen induces expression of tdTomato specifically in astrocytes. We injected these mice at P60 with a low dose of tamoxifen in order to sparsely label astrocytes (Fig. 1b). We then examined whether astrocytes in different layers manifested distinct morphological features, including differences in territorial volume and cell orientation. Quantification of territorial volume revealed that layer I astrocytes occupied a significantly smaller volume compared with astrocytes in layers II/III–VI, whereas layer II/III astrocytes occupied a larger volume than did astrocytes in layers IV–VI. The territorial volumes of astrocytes in layers IV, V, and VI were similar (Fig. 1c).

Quantification of 3D-cell orientation also revealed that the average orientation angle relative to the pial surface of layer II/III astrocytes was closer to 90°, whereas that of layer VI astrocytes was closer to 0°, emphasizing the radial orientation of layer II/III astrocytes and the tangential orientation of layer VI astrocytes (Fig. 1d). We also observed similar layer-specific differences in the orientation of sparsely labeled astrocytes in another reporter mouse line, Sox9-CreER^T2;CAG-CAT-eGFP at P120 (Supplementary Fig. 1a–d). We confirmed this finding by immunostaining for the astrocyte markers GFAP (glial fibrillary acidic protein) (Supplementary Fig. 1e).

Furthermore, 3D Sholl analysis of microtubule-containing processes automatically extracted from a tdTomato-labeled region of interest (ROI) showed that the extent of process arborization (at a distance of >30 μm from the nucleus) was greater for layer II/III astrocytes than for layer VI astrocytes (Fig. 1e).

**Unbiased cluster analysis of astrocyte morphologies**. Given that astrocytes in different layers appeared to have distinct morphologies, we examined further the relation between morphological properties and laminar position in an unbiased manner. We thus classified neocortical astrocytes into subpopulation clusters based on morphological features and then examined how the clusters were distributed across neocortical layers. We first measured 24 morphometric parameters for sparsely labeled neocortical astrocytes in Glast-EMTB-GFP;Glast-CreER^T2;Rosa-CAG-LSL-tdTomato mice at P65 ($n = 116$ cells from five mice) (Supplementary Table 1). According to previous clustering studies[20,21], we applied a multimodal index (MMI) of parameters for cluster analysis of >0.55, with the following parameters then being used as clustering criteria: elongation, flatness, XY, relative XZ, and relative YZ.

The t-distributed stochastic neighbor embedding (t-SNE)[22] analysis, which visualizes the relative distribution of samples, showed that neocortical astrocytes largely separated into two

**Fig. 1** Three-dimensional morphology of astrocytes in layers I–VI of the mouse somatosensory cortex as revealed by visualization of microtubule and whole-cell structures. **a** 3D reconstruction of astrocyte morphology in the Glast-EMTB-GFP mouse brain (P60) rendered transparent by the CUBIC technique. 3D structures and traces are shown in the left panels, and individual traces in the right. CC, corpus callosum. Scale bars, 250 μm (left panels), 50 μm (right panels). **b** Representative confocal image of sparsely labeled astrocytes in Glast-EMTB-GFP;Glast-CreER^T2;Rosa-CAG-LSL-tdTomato mice (P65) injected with a low dose of tamoxifen at P60 (left), and representative 3D projection images of astrocytes in each layer (right). Arrowhead indicates GFP+tdTomato+ cell. Scale bars, 250 μm (left panel), 50 μm (right panels). **c, d** Quantification of territorial volume (**c**) and the angle of orientation relative to the brain surface (**d**) for astrocytes in each layer. The data are shown for 116 cells from five brains, with the red bars indicating median values. *$P < 0.05$, **$P < 0.01$, ***$P < 0.001$ (one-way ANOVA followed by Bonferroni's test). **e** 3D Sholl analysis for microtubule structure of sparsely labeled individual astrocytes in layers II/III and VI of Glast-EMTB-GFP;Glast-CreER^T2;Rosa-CAG-LSL-tdTomato mice injected with a low dose of tamoxifen at P60. The data are means (layer II/III astrocytes, $n = 27$ cells from five brains; layer VI astrocytes, $n = 16$ cells from five brains). *$P < 0.05$, **$P < 0.01$, ***$P < 0.001$ versus corresponding values for layer VI astrocytes (two-way ANOVA followed by Bonferroni's test)

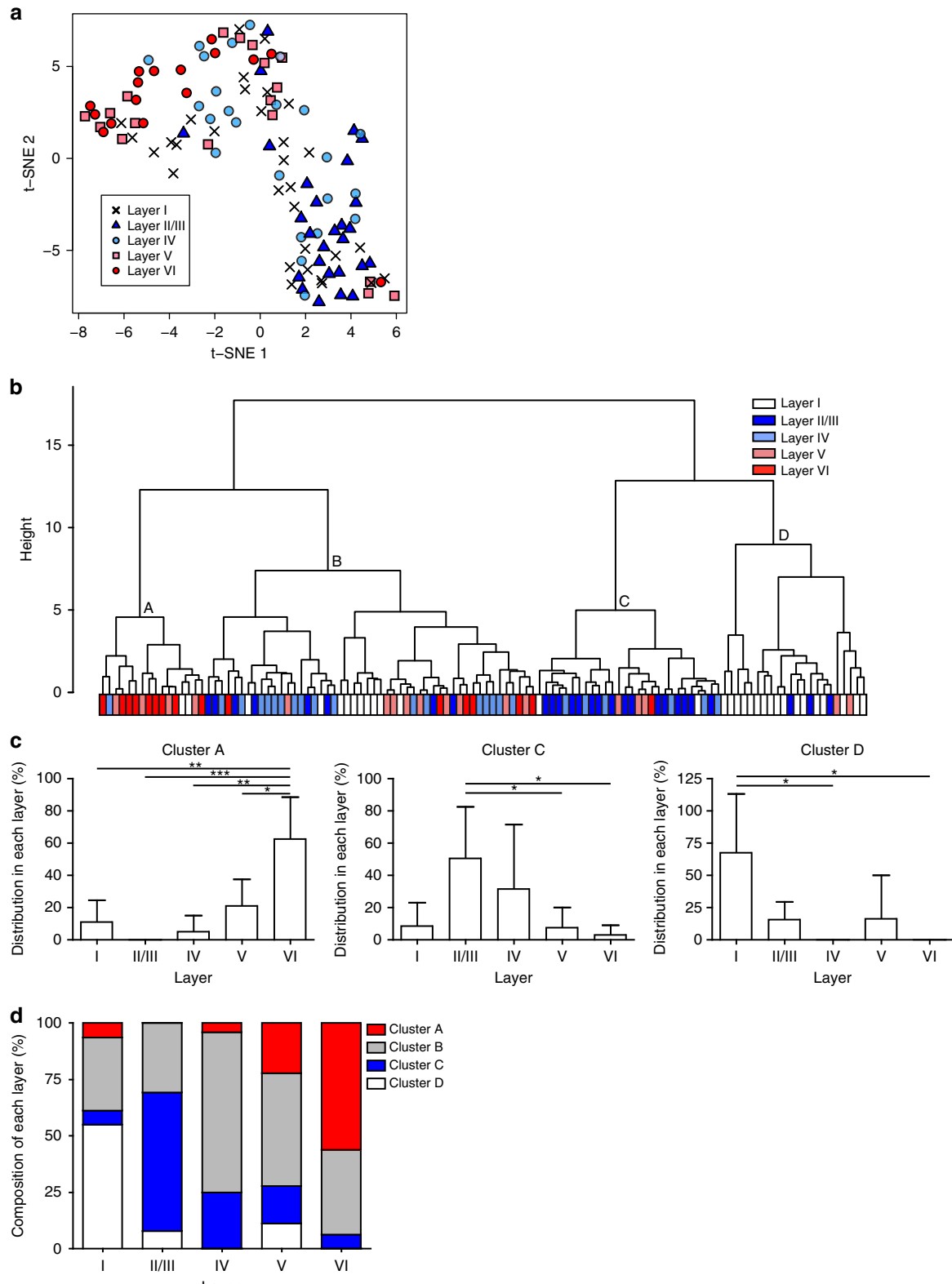

**Fig. 2** Cluster analysis based on morphological features of neocortical astrocytes. **a** t-SNE analysis showing separation of neocortical astrocytes into two groups, with most cells in layer II/III and those in layers V and VI localizing to different groups. **b** Hierarchical clustering showing that neocortical astrocytes can be separated into four clusters designated A through D, with cluster A being enriched in astrocytes of layers V and VI, cluster C in those of layer II/III, and cluster D in those of layer I. **c** Percentage of astrocytes in each cluster located in the different neocortical layers (cluster A, $n = 16$; cluster C, $n = 28$; cluster D, $n = 22$). Data are means ± s.d. for at least three mice. *$P < 0.05$, **$P < 0.01$, ***$P < 0.001$ (one-way ANOVA followed by Bonferroni's test). **d** Percentage of astrocytes in each layer belonging to clusters A through D (layer I, $n = 31$; layer II/III, $n = 27$; layer IV, $n = 24$; layer V, $n = 18$; layer VI, $n = 16$). Two-sided Fisher's exact test, $P = 8.439 \times 10^{-12}$

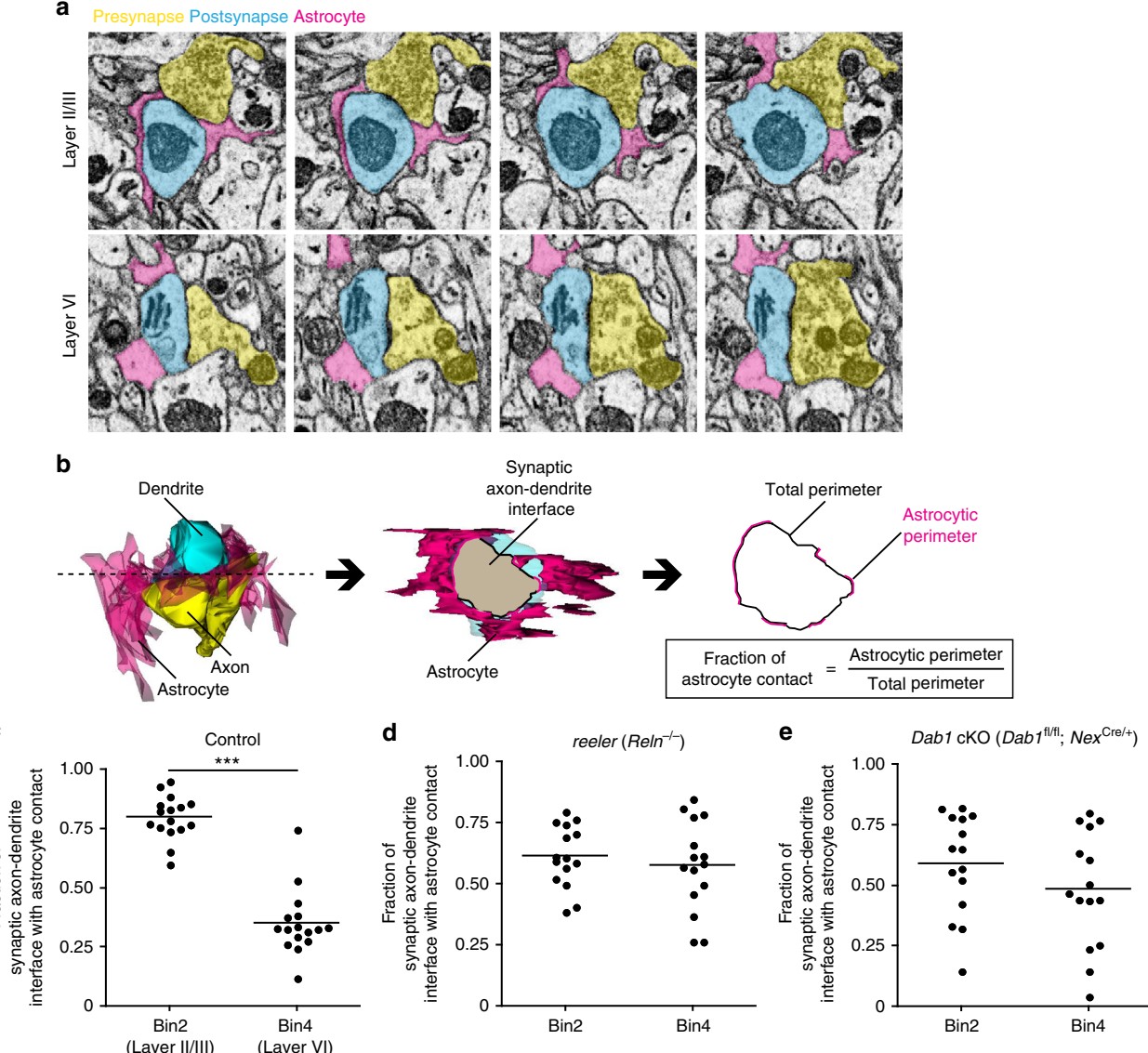

**Fig. 3** Difference in astrocyte-synapse structural interactions in layers II/III and VI. **a** Series of SBF-SEM images acquired from coronal sections of the brain of an adult wild-type mouse at ×4730 magnification. **b** Procedure for quantification of astrocyte ensheathment of synapses. Serial SEM images were reconstructed to measure the perimeter of synaptic axon-dendrite interface and the astrocytic perimeter. **c–e** Quantification of astrocyte ensheathment of synapses in bin2 and bin4 ($n = 16$ synapses in each bin for control (**c**) and $n = 15$ synapses in each bin for *reeler* (**d**) and *Dab1* cKO (**e**) mice). Astrocyte position in the cortical layers is expressed as relative distance from the corpus callosum (CC) to the pia: bin2, 0.65–0.9; and bin4, 0–0.25. Horizontal bars represent mean values. ***$P < 0.001$ (Welch's *t*-test)

groups (Fig. 2a). Notably, most layer II/III astrocytes and most layer V or VI astrocytes separated into the different groups. This finding thus supported the notion that astrocytes in different layers possess distinct morphological features.

Ward's hierarchical clustering analysis[23] revealed that neocortical astrocytes can be classified into four clusters (clusters A through D in Fig. 2b). We took into account the total intracluster distance[24] and silhouette analysis[25] to determine the optimal number of clusters (Supplementary Fig. 2a, b). Astrocytes in each cluster showed different morphological features. For example, astrocytes in cluster D were significantly flatter than those in the other three clusters (Supplementary Fig. 2c). In addition, most astrocytes in cluster A showed a tangential orientation, whereas most of those in clusters C and D showed a radial orientation (Supplementary Fig. 2d). To confirm the morphological differences among astrocytes in the different clusters, we performed 3D Sholl analysis of process arborization, a morphometric parameter

not used as a clustering criterion. Such analysis revealed that the extent of process arborization was greater for astrocytes in clusters B and C than for those in clusters A and D (Supplementary Fig. 2e).

Importantly, astrocytes with distinct morphologies (those in the different clusters) were differentially distributed among cortical layers, with those of cluster A and those of cluster C being significantly enriched in layer VI and layer II/III, respectively (62.50% and 50.57%, respectively) (Fig. 2c). Most astrocytes of cluster D were located in layer I (67.71%) (Fig. 2c), showing that astrocytes in this layer differ morphologically from those in other layers. Thus, layer I was composed mostly of astrocytes in cluster D (54.8%), layer II/III of astrocytes in cluster C (61.5%), layer IV of astrocytes in cluster B (70.8%), layer V of astrocytes in cluster B (50.0%), and layer VI of astrocytes in cluster A (56.3%) (two-sided Fisher's exact test, $P = 8.439 \times 10^{-12}$) (Fig. 2d). Our findings show that, in addition to layer I

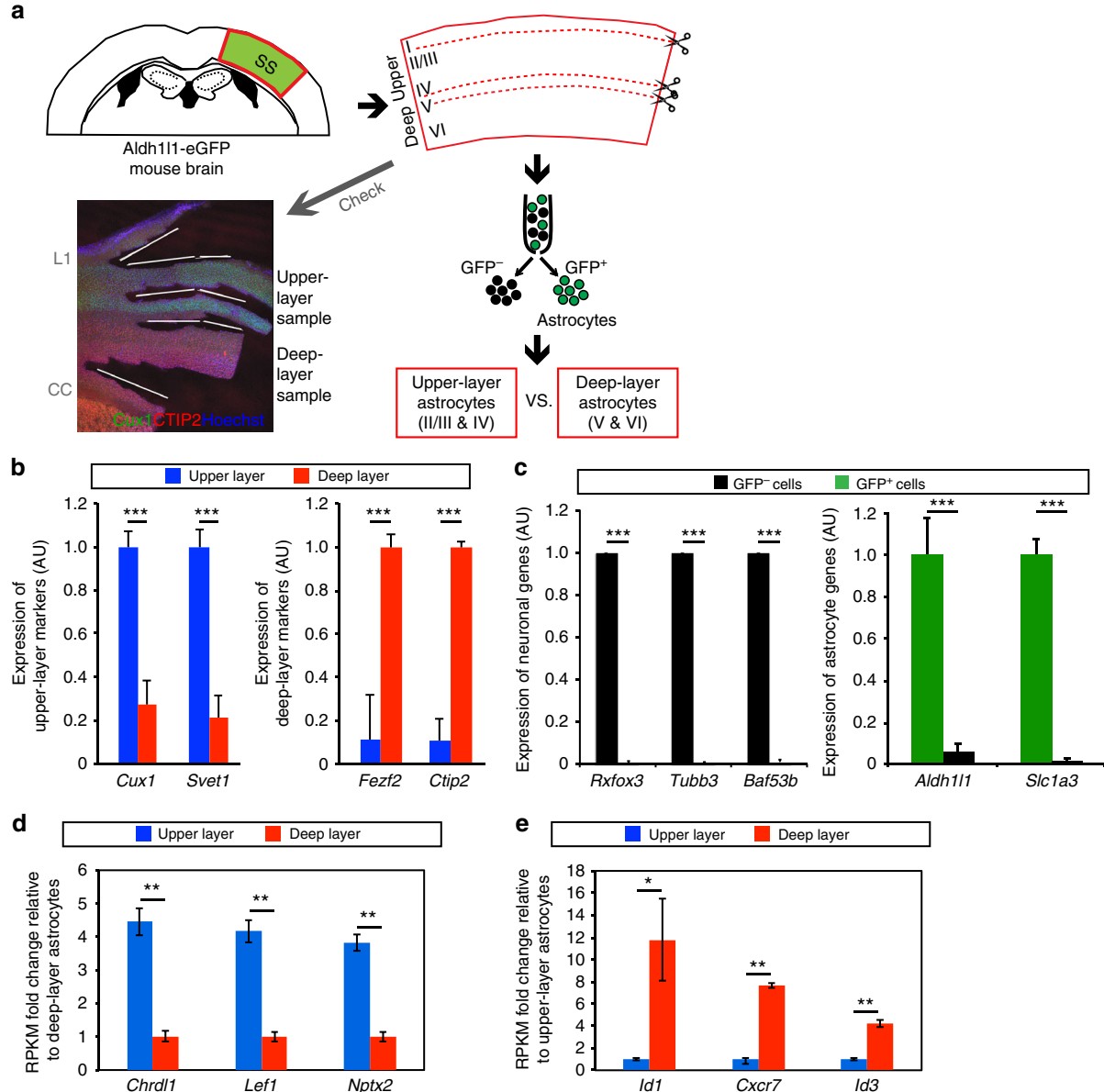

**Fig. 4** Layer-specific molecular expression in neocortical astrocytes. **a** Procedure for the isolation of upper-layer (ULAs) and deep-layer (DLAs) astrocytes for comparison of their gene expression levels. Layer I (L1), the boundary between layers IV and V, and the corpus callosum (CC) are discarded from the somatosensory (SS) area of the brain of young adult Aldh1l1-eGFP transgenic mice in which astrocytes are labeled with GFP and can therefore be isolated by FACS. The fluorescence image is of a coronal section stained with antibodies to Cux1 and to CTIP2, as well as with Hoechst 33342. **b** RT-qPCR analysis of upper-layer (*Cux1*, *Svet1*) and deep-layer (*Fezf2*, *Ctip2*) neuronal marker genes in upper-layer and deep-layer tissue samples. **c** RT-qPCR analysis of neuronal (left) and astrocytic (right) marker genes in isolated GFP+ or GFP− cells. **d**, **e** RPKM fold change relative to deep-layer (**d**) and upper-layer (**e**) in expression analyzed by RNA-seq (n = 3 brains). Top 3 upper-layer enriched and deep-layer enriched genes are shown in **d**, **e**, respectively. See Supplementary Data 1 and 2 for full gene lists of upper-layer and deep-layer enriched genes, respectively. Data are means ± s.d. for three brains. *P < 0.05, **P < 0.01, ***P < 0.001 (paired two-tailed Student's *t*-test)

astrocytes, those in layers II–VI also exhibit layer-specific morphological differences.

**Synaptic interaction of astrocytes in different layers**. To investigate whether astrocytes in each cluster classified according to the morphological features in Supplementary Table 1—in particular, those of clusters A and C—might also possess different functional properties, we examined their structural interactions with synapses by serial scanning electron microscopy (SEM)[26]. According to the tripartite synapse concept[1,27], perisynaptic astrocytes are present along with the presynaptic and

postsynaptic neurons. The intricate ramifications of astrocytes allow them to tightly enwrap the synaptic terminal and to modulate synaptic processes[28–30]. We hypothesized that astrocytes of cluster C, with their more pronounced arborization (Supplementary Fig. 2e), might interact to a greater extent with neighboring synapses compared with those of cluster A. We therefore observed the structural interaction of perisynaptic astrocytes in layers II/III and VI, in which astrocytes of cluster C and cluster A are respectively enriched (Fig. 2c). The 3D reconstruction of SEM images obtained with a focused ion beam (FIB) indeed showed that the extent of ensheathment of synaptic clefts by astrocytes in layer II/III was greater than that by those in layer

**Table 1 Upper-layer enriched astrocytic genes**

| Gene symbol | RPKM | | Fold change (upper/deep) | | |
| --- | --- | --- | --- | --- | --- |
| | Upper | Deep | | P value | Known functions in astrocytes (Refs) |
| Chrdl1 | 89.5 | 20.1 | 4.5 | 0.0038 | |
| Lef1 | 19.4 | 4.6 | 4.3 | 0.0074 | |
| Fmo1 | 32.5 | 12.2 | 2.7 | 0.0064 | |
| Ccdc80 | 24.1 | 17.3 | 1.4 | 0.019 | |
| Dio2 | 39.7 | 28.9 | 1.4 | 0.0091 | Regulation of fatty acid oxidation level ([34]) |
| Slc1a3 | 2538.5 | 1893.0 | 1.3 | 0.0056 | Regulation of synaptic activity ([39]) |
| Slc15a2 | 137.8 | 103.4 | 1.3 | 0.0097 | Regulation of peptide transport ([35]) |
| Fgfr3 | 302.3 | 227.0 | 1.3 | 0.0087 | Negative regulator of Gfap expression ([40]) |
| Cyp4f15 | 271.0 | 205.8 | 1.3 | 0.028 | |
| Mertk | 227.9 | 178.1 | 1.3 | 0.024 | Regulation of synapse elimination ([36]) |
| Vcam1 | 193.9 | 156.4 | 1.2 | 0.032 | |
| Gja1 | 2230.3 | 1847.3 | 1.2 | 0.027 | Regulation of cell survival ([37]) |

Average RPKM and fold changes are shown (n = 3 brains). P value, paired t-test
See Supplementary Data 1 and 3 for full gene lists of upper-layer enriched genes

**Table 2 Deep-layer enriched astrocytic genes**

| Gene symbol | RPKM | | Fold change (deep/upper) | | |
| --- | --- | --- | --- | --- | --- |
| | Upper | Deep | | P value | Known functions in astrocytes (Refs) |
| Id1 | 3.7 | 43.9 | 11.6 | 0.035 | |
| Cxcr7 | 0.7 | 5.6 | 8.9 | 0.0017 | Regulation of proliferation ([43]) |
| Id3 | 126.2 | 529.8 | 4.2 | 0.0030 | |
| Fgfbp3 | 3.6 | 9.0 | 2.9 | 0.017 | |
| Sparc | 20.4 | 40.9 | 2.0 | 0.013 | Negative regulation of synapse formation ([41,44]) |
| Gfap | 11.7 | 23.5 | 2.0 | 0.045 | Regulation of neuron-astrocyte communication ([42]) |

Average RPKM and fold changes are shown (n = 3 brains). P value, paired t-test
See Supplementary Data 2 and 3 for full gene lists of deep-layer enriched genes

VI (Fig. 3a–c), suggesting that astrocytes in different layers show differences not only in morphology but also in synaptic interaction.

**Layer-specific molecular expression in cortical astrocytes.** We next questioned whether neocortical astrocytes also show layer-specific molecular differences by comparing the gene expression profiles of astrocytes in different layers. Given that most astrocytes of clusters A and C reside in layers V and VI and layers II–IV, respectively (Fig. 2c), we compared astrocytes of these layers, hereafter referred to as deep-layer astrocytes (DLAs) and upper-layer astrocytes (ULAs), respectively. With the use of fluorescence-activated cell sorting (FACS), we prepared ULAs and DLAs from the corresponding dissected layers of the somatosensory cortex of Aldh1l1-eGFP mice[31] in which all astrocytes are expected to be labeled with GFP. The meninges, layer I, and the corpus callosum were removed from upper- and deep-layer tissue samples. In addition, parts of layers IV and V were lost during separation of these layers in such a way as to prevent cross-contamination between the upper- and deep-layer samples (Fig. 4a).

Immunostaining, as well as reverse transcription (RT) and quantitative polymerase chain reaction (qPCR) analysis of upper-layer and deep-layer neuronal markers (Cux1 and Ctip2 in both analyses, and Svet1 and Fezf2 in RT-qPCR analysis for further confirmation) verified the effectiveness of the sample preparation procedure (Fig. 4a, b). RT-qPCR analysis of neuronal and astrocytic markers showed that astrocytes were appropriately isolated as a GFP-positive cell population with negligible contamination by neurons (Fig. 4c). We next compared gene expression patterns between ULAs and DLAs by RNA-sequencing (RNA-seq) analysis (n = 3). We identified genes that were expressed preferentially in either ULAs or DLAs (Fig. 4d, e, Tables 1 and 2, Supplementary Data 1 and 2). Although, most genes known to be preferentially expressed in astrocytes (astrocyte-enriched genes)[16,31–33] were equally expressed between ULAs and DLAs (Supplementary Data 3), some such genes (astrocyte-enriched genes or genes with known function in astrocytes)[34–44] were significantly enriched in ULAs or DLAs (Tables 1 and 2). These astrocytic genes include extracellular secreted proteins (Chrdl1, Sparc), transporters (Slc1a3(GLAST), Slc15a2), enzymes (Fmo1, Dio2), receptors/membrane proteins (Fgfr3, Mertk, Vcam1, Gja1(Connexin43), and Cxcr7), cytoskeletal protein (Gfap) and transcription factors (Id1, Id3, and Lef1). Together, these results showed that ULAs and DLAs differ in their molecular expression patterns. RNA-seq data has been deposited to the GEO database (accession code GSE111916).

To confirm layer-specific molecular expression in neocortical astrocytes, we then examined the expression of Lef1 and Id1 proteins, the mRNA for which was more abundant in ULAs and in DLAs, respectively, according to our RNA-seq analysis (Supplementary Data 1 and 2). We found that the intensity of Lef1 immunostaining was higher in ULAs than in DLAs (Fig. 5a, c), whereas that of Id1 immunostaining was higher in DLAs (especially those located in layer VI), as well as in layer I astrocytes than in ULAs located in layers II–IV (Fig. 6a, c). The percentages of Lef1- or Id1-positive astrocytes among astrocytes in each area (bins 1–4) indicate the enrichment of Lef1 protein in

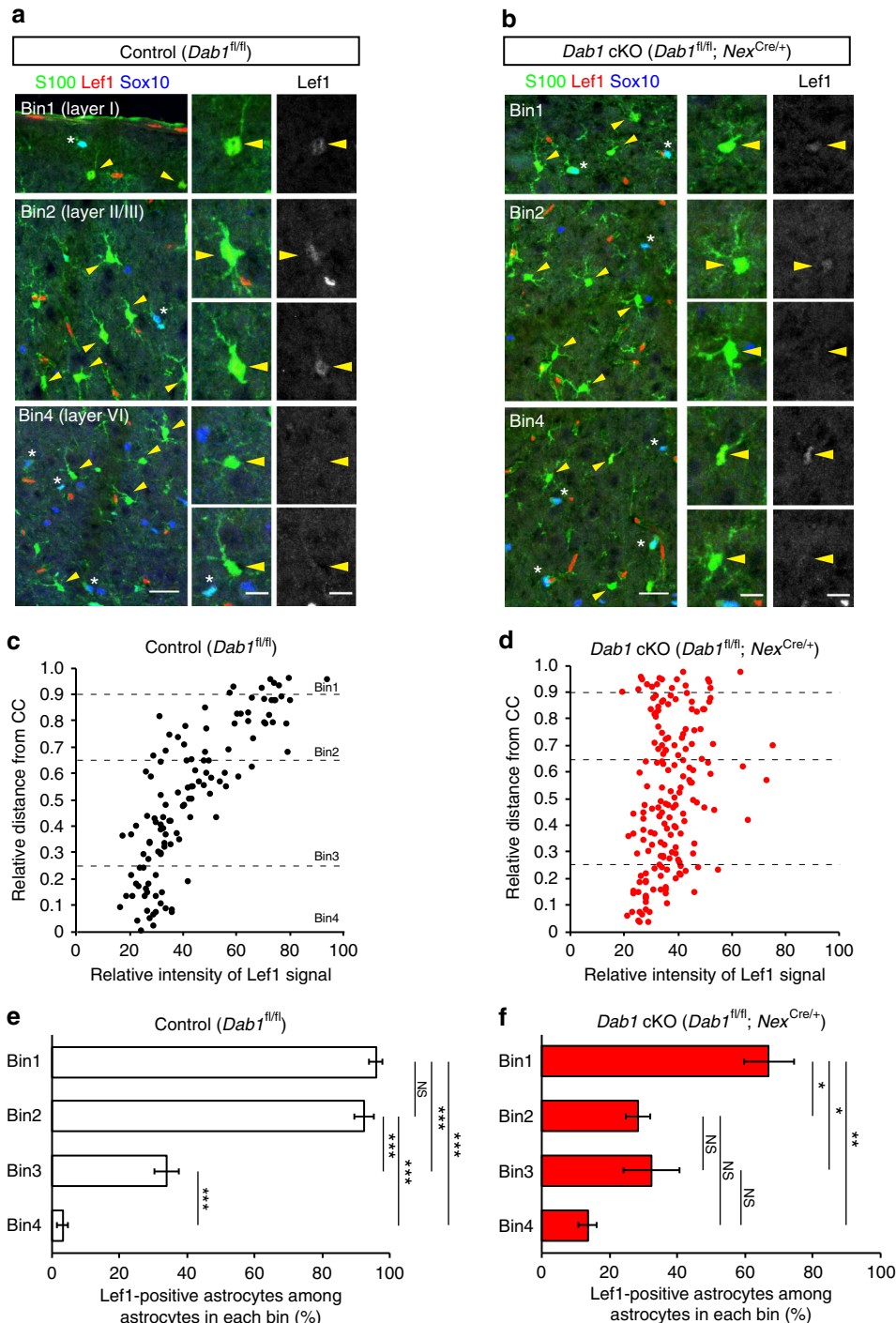

**Fig. 5** Layer-specific expression of Lef1 in neocortical astrocytes and its disruption in *Dab1* cKO mice. **a, b** Immunofluorescence staining for S100, Sox10, and Lef1 in coronal sections of control (*Dab1*fl/fl) (**a**) and *Dab1* cKO (*Dab1*fl/fl;*Nex*Cre/+) (**b**) mice at P30. Astrocytes were identified as S100-positive, Sox10-negative cells (arrowheads); asterisks indicate double-positive cells. Individual astrocytes and Lef1 signals are shown in the right panels at higher magnification. Scale bars, 25 μm (left panels) and 10 μm (right panels). **c, d** Representative data of Lef1 signal intensitiesin the nucleus of individual astrocytes in a coronal section from a control (**c**) and a Dab1 cKO (**d**) mouse. Astrocyte position in the cortical layers is expressed as relative distance from the corpus callosum (CC) to the pia, with dashed lines indicating the boundaries between bins: bin1, 0.9–1.0; bin2, 0.65–0.9; bin3, 0.25–0.65; and bin4, 0–0.25. **e, f** Percentage of Lef1-positive astrocytes among total astrocytes in each bin of the neocortex of control (**e**) and *Dab1* cKO (**f**) mice (*n* = 3 mice). Perivascular astrocytes are excluded from quantification. Data are means ± s.e.m. from three mice of each genotype. **\*\*P < 0.01; \*\*\*P < 0.001; NS, not significant (one-way ANOVA followed by Bonferroni's test)

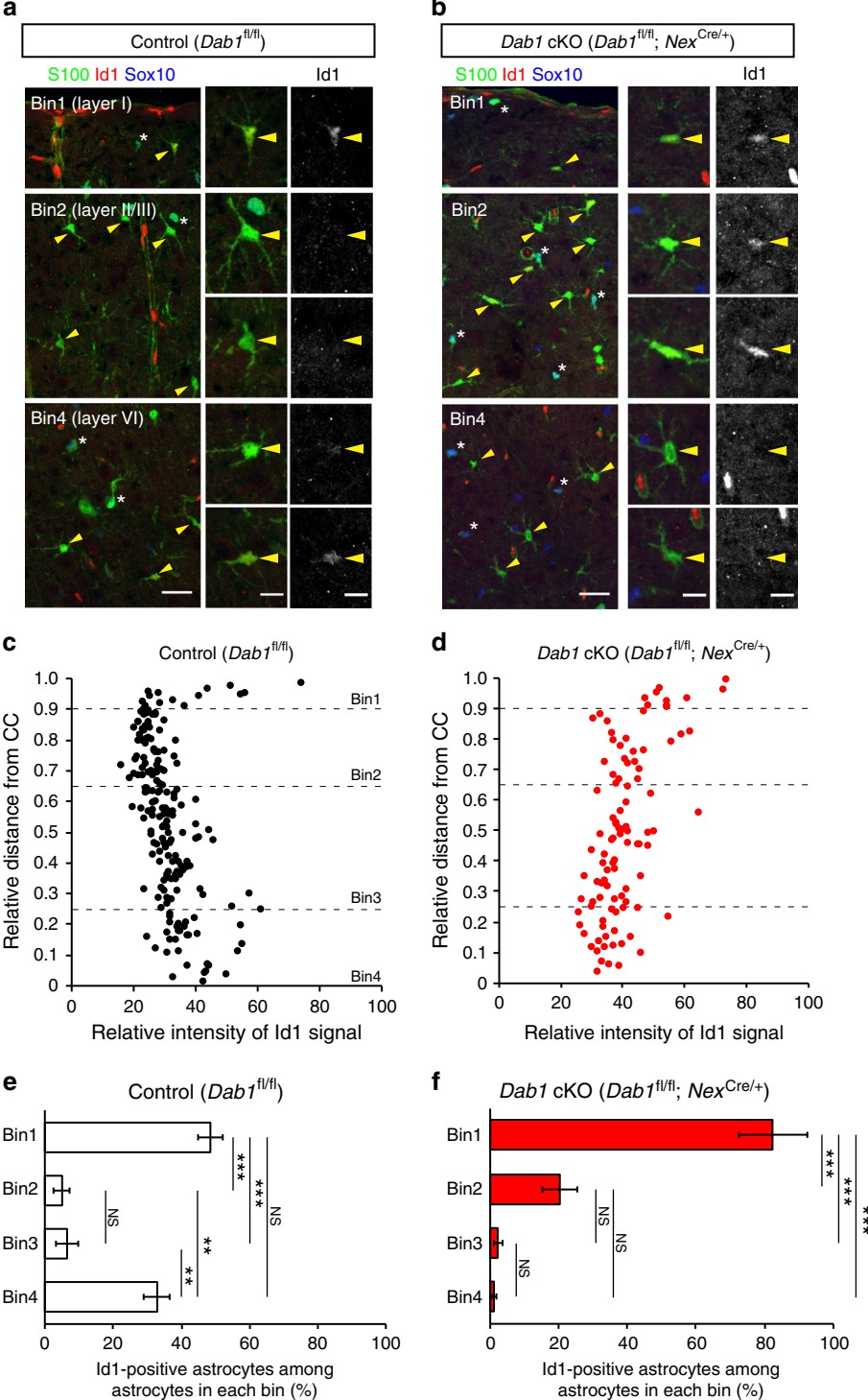

**Fig. 6** Layer-specific expression of Id1 in neocortical astrocytes and its disruption in *Dab1* cKO mice. **a**, **b** Immunofluorescence staining for S100, Sox10, and Id1 in coronal sections of control (*Dab1*[fl/fl]) (**a**) and *Dab1* cKO (*Dab1*[fl/fl];*Nex*[Cre/+]) (**b**) mice at P30. Astrocytes were identified as S100-positive, Sox10-negative cells (arrowheads); asterisks indicate double-positive cells. Individual astrocytes and Id1 signals are shown in the right panels at higher magnification. Scale bars, 25 μm (left panels) and 10 μm (right panels). **c**, **d** Representative data of Id1 signal intensities in the nucleus of individual astrocytes in a coronal section from a control (**c**) and a Dab1 cKO (**d**) mouse. Astrocyte position in the cortical layers is expressed as relative distance from the corpus callosum (CC) to the pia, with dashed lines indicating the boundaries between bins: bin1, 0.9–1.0; bin2, 0.65–0.9; bin3, 0.25–0.65; and bin4, 0–0.25. **e**, **f** Percentage of Id1-positive astrocytes among total astrocytes in each bin of the neocortex of control (**e**) and *Dab1* cKO (**f**) mice (*n* = 3 mice). Perivascular astrocytes are excluded from quantification. Data are means ± s.e.m. from three mice of each genotype. **P < 0.01; NS, not significant (one-way ANOVA followed by Bonferroni's test)

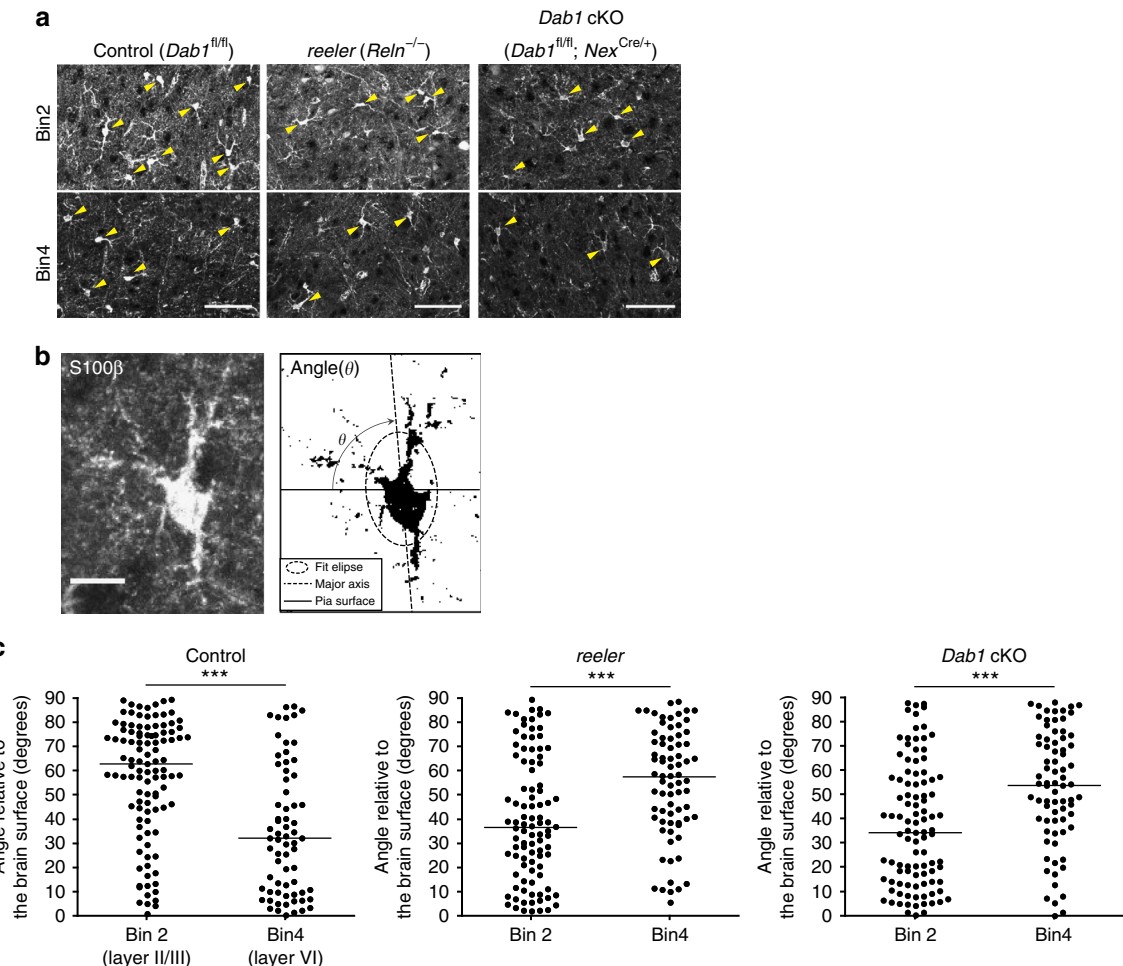

**Fig. 7** Disruption of layer-specific astrocyte orientation in *reeler* and *Dab1* cKO mice. **a** Immunofluorescence staining for S100β in bin2 (layer II/III) and bin4 (layer VI) of coronal brain sections from control (*Dab1*$^{fl/fl}$), Dab1 cKO (*Dab1*$^{fl/fl}$;*Nex*$^{Cre/+}$), and *reeler* (*Reln*$^{-/-}$) mice at P60–P70. Arrowheads indicate S100β-positive astrocytes. Scale bars, 50 μm. **b** Schematic for measurement of astrocyte orientation relative to the brain surface based on S100β immunostaining. Scale bar, 10 μm. **c** Quantification of the orientation angle of astrocytes (S100β-positive, Sox10-negative cells) in bin2 and bin4 of control, *Dab1* cKO, and *reeler* mice (*n* = 3 mice) at P60–P70. Horizontal bars indicate median values. Control, *n* = 106 and 68 cells; *Dab1* cKO, *n* = 100 and 76 cells; *reeler*, *n* = 96 and 74 cells for bin2 and bin4, respectively. ***$P < 0.001$ (Welch's *t*-test)

layers I and II/III astrocytes (in bins 1 and 2) (Fig. 5e) and that of Id1 protein in layers I and VI astrocytes (in bins 1 and 4) (Fig. 6e). Moreover, the immunostaining intensities of high mobility group nucleosome-binding 1–3 (HMGN1–3) and Zbtb20, which play key roles in astrocyte fate determination and differentiation[45,46], were higher in ULAs than in DLAs (Supplementary Fig. 3). Together, our results thus revealed expression differences between ULAs and DLAs at both mRNA and protein levels and are therefore indicative of layer-specific differences in molecular expression among neocortical astrocytes.

**Neurons regulate layer-specific astrocytic properties**. We next asked whether neuronal layers are required for the establishment of the observed layer-specific astrocytic properties during development. We first examined *reeler* mice, which are deficient in the guidance protein Reelin[47,48]. Reelin is secreted predominantly from Cajal-Retzius cells that reside in layer I and regulates the migration of cortical neurons. The loss of Reelin results in defects in neuronal laminar structure[49], which we confirmed by detection of an inside-out inversion of neurons positive for Cux1 or Ctip2 (Supplementary Fig. 4a, b). We found that the layer-specific

pattern of astrocyte orientation detected by S100β immunostaining, which is consistent with 3D and 2D cell orientation analysis (Fig. 1, Supplementary Fig. 1), was also inverted in *reeler* mice (Fig. 7). This result showed that the layer-specific orientation of neocortical astrocytes was dependent on Reelin.

However, Reelin might directly affect astrocytes and neural progenitors in addition to migrating neurons, it was desirable to perturb neuronal layers by manipulating only neurons. We therefore examined *Dab1*$^{fl/fl}$;*Nex*$^{Cre/+}$ mice in which floxed *Dab1* alleles are deleted specifically in neurons. Dab1 is an essential mediator of Reelin-induced neuronal migration and, indeed, the *Dab1* cKO mice manifest an inside-out inversion of cortical neuronal layers (Supplementary Fig. 4a, c)[50]. Importantly, we found that the layer-specific pattern of astrocyte orientation revealed by S100β immunostaining was inverted in *Dab1*$^{fl/fl}$;*Nex*$^{Cre/+}$ mice (Fig. 7), similar to the phenotype found in *reeler* mice. Moreover, the greater extent of arborization within a coronal plane which contained the nucleus for layer II/III astrocytes compared with layer VI astrocytes was also no longer apparent in *Dab1*$^{fl/fl}$;*Nex*$^{Cre/+}$ mice (Fig. 8). Furthermore, the enrichment of Lef1 protein in layer II/III astrocytes (bin2) over layer VI astrocytes (bin4), as well as that of Id1 protein in layer VI

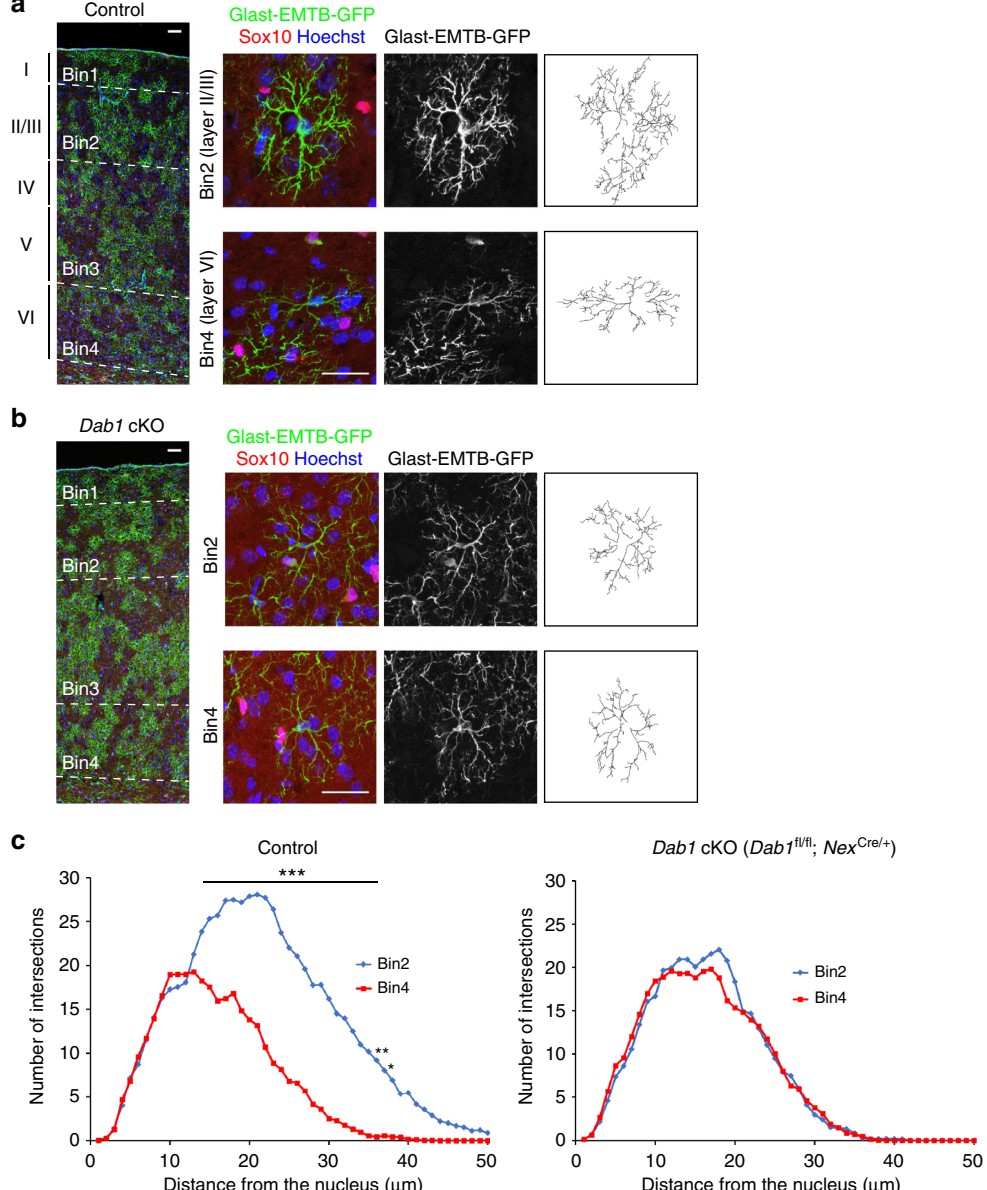

**Fig. 8** Disruption of layer-specific astrocyte arborization in *Dab1* cKO mice. **a**, **b** Immunofluorescence staining for GFP and Sox10 in coronal sections of control (Glast-EMTB-GFP;*Dab1*fl/fl or Glast-EMTB-GFP) (**a**) and *Dab1* cKO (Glast-EMTB-GFP;*Dab1*fl/fl;*Nex*Cre/+) (**b**) mice at P30 (left panels). Nuclei were stained with Hoechst 33342. Scale bars, 50 μm. Right panels show representative magnified images of astrocytes and their traced processes in bin2 (layer II/III) and bin4 (layer VI). Scale bars, 25 μm. **c** 2D Sholl analysis for microtubule structure of individual astrocytes in coronal brain sections. Astrocytes were randomly selected for analysis. Data are means for 15 cells in bin2 and 18 cells in bin4 for control mice and for 18 cells in bin2 and 17 cells in bin4 for *Dab1* cKO mice. Three brains of each genotype were analyzed. *$P < 0.05$, **$P < 0.01$, ***$P < 0.001$ (two-way ANOVA followed by Bonferroni's test)

astrocytes (bin4) over layer II/III astrocytes (bin2) were lost in *Dab1*fl/fl;*Nex*Cre/+ mice (Figs. 5e, f and 6e, f, S5a–d). Collectively, these results indicated that neuronal layers play an essential role in the establishment or maintenance of layer-specific properties of neocortical astrocytes.

Finally, we investigated whether neuronal laminar defects might affect astrocyte ensheathment of synapses. Serial SEM images showed that the greater extent of astrocyte ensheathment of synaptic clefts in layer II/III compared with that in layer VI observed in control mice was not apparent in *reeler* mice or *Dab1* cKO mice (Fig. 3c–e). This finding thus suggested that layer-specific synaptic ensheathment by astrocytes also depends on neuronal layers.

## Discussion

Although, neocortical astrocytes have long been thought to largely comprise a nondiverse population of protoplasmic astrocytes distributed throughout cortical layers, we have now shown that neocortical astrocytes in layers II–VI exhibit layer-specific properties in terms of morphology, structural interactions with synapses, and molecular expression. Morphological differences in neocortical astrocytes were consistently observed in different experimental approaches: visualization of intermediate filament structure by GFAP staining, S100β staining, visualization of microtubule structure in Glast-EMTB-GFP transgenic mice, and visualization of whole-cell shape in either Sox9-CreER^T2;CAG-CAT-eGFP or Glast-CreER^T2;Rosa-CAG-LSL-tdTomato reporter

mice. Similarly, an unbiased cluster analysis revealed that astrocytes with different morphological features are distributed in a layer-specific manner. Furthermore, our morphological observations uncovered differences between astrocytes in layer I and those in other layers, consistent with previous studies showing differences in $Ca^{2+}$ dynamics and gene expression between these cells[11,13].

Whether astrocytes with different morphologies have different functions or engage in different interactions with neighboring neurons or synapses remains to be further investigated, however. The radial and tangential elongation of astrocytes located in upper and deep layers, respectively, may suggest the existence of radial and tangential domains of synapses regulated by these cells. Similarly, the greater territorial volume of layer II/III astrocytes may indicate that these cells have a larger astrocytic domain and constitute a larger glio-neuronal unit, which was recently proposed as an elementary (astrocyte-governing) unit of the brain that monitors, integrates, and potentially modifies the activity of a contiguous set of synapses[51]. In addition, the difference in process complexity between ULAs and DLAs might give rise to a difference in structural interactions with synapses. It has been shown that connexin 30 knockout leads to morphological changes (increased domain area, elongated processes, and enhanced ramification) of astrocytes, increased intrusion of their processes into synaptic clefts, enhanced astroglial glutamate clearance, a consequent decrease in excitatory synaptic transmission and impairment of synaptic plasticity[52]. Our SEM observations indicated that perisynaptic astrocytes provide more synapse ensheathment in layer II/III, where most astrocytes show a greater extent of process arborization compared with those in layer VI. These findings suggest that astrocytes in different layers with distinct morphologies may interact differentially with neighboring synapses, possibly giving rise to functional heterogeneity in modulation of glutamate clearance and synaptic plasticity.

We also detected differences in gene and protein expression patterns between ULAs and DLAs. RNA-seq and immunostaining data revealed that ULAs and DLAs differentially express various molecules related to synaptic regulation, morphogenesis, and metabolism. In addition, we detected differences in expression of the transcription factors Zbtb20 and HMGN1–3, which play important roles in astrocyte fate determination and differentiation[45,46]. Astrocyte subpopulations with different molecular expression patterns have recently been shown to be heterogeneous both functionally with regard to regulation of axon projection in the spinal cord and synaptogenesis in vitro as well as in their contribution to pathological states (8, 16). It will be of interest to determine whether the molecules differentially expressed by ULAs and DLAs contribute to the functional heterogeneity of neocortical astrocytes. For instance, *Mertk*, which mediates synapse elimination[36,44], and *Sparc*, which encodes an inhibitor of excitatory synapse formation[41,44], are expressed at a higher level in ULAs and DLAs, respectively. This differential expression possibly influences neighboring synapses in a layer-specific manner. Moreover, we found that *Chrdl1*, which encodes an inhibitor of bone morphogenetic protein (BMP) signaling, is expressed at a higher level in ULAs than in DLAs, whereas Id family transcription factors, which are downstream effectors of BMP signaling, are expressed at a higher level in DLAs. These findings are suggestive of a gradient of BMP signaling in astrocytes across the cortical layers. Together, these results show that astrocytes in distinct layers differentially express various molecules, with such differences possibly giving rise to functional heterogeneity of astrocytes in the mouse neocortex.

How are these layer-specific properties of astrocytes established during development? It is possible that different progenitors give

rise to astrocytes that reside in different layers. However, lineage-tracing experiments have indicated that clones of astrocytes in layers II/III–VI show an extensive distribution among several cortical layers[12,53], suggesting that ULAs and DLAs are derived from common progenitors. We therefore considered that the different properties of astrocyte subtypes might be determined either by intrinsic programs within the progenitors or by extrinsic signals derived from neighboring cells including neurons. We therefore investigated whether the layer-specific properties of neocortical astrocytes are affected in *reeler* mice or in *Dab1* cKO mice in which the laminar structure of the cerebral cortex is defective. We found that layer-specific expression of Lef1 and Id1, as well as layer-specific differences in astrocyte morphology and synaptic ensheathment were lost in these mice. These results thus indicate that neuronal layers are a prerequisite for establishment of the layer-specific properties of cortical astrocytes, which may answer in part the long-standing question of why neurons are produced before astrocytes during development.

In this study, we have uncovered the existence of layer-specific properties of neocortical astrocytes corresponding to the laminar organization of neurons. In addition, the results from *Dab1* cKO mice suggest that astrocytes diversify following nonuniform extrinsic signals originated from neurons. Our findings thus provide a basis for future studies regarding the diversification and functions of specified glio-neuronal networks.

## Methods

**Animals**. Aldh1l1-eGFP (MMRRC stock #11015) transgenic mice were obtained from the University of California at Davis. Glast-EMTB-GFP transgenic mice and Glast-CreER[T2] knockin mice[54] were kindly provided by E. S. Anton (University of North Carolina School of Medicine) and M. Götz (Munich Center for Neurosciences), respectively. Rosa-CAG-LSL-tdTomato mice[55] were obtained from The Jackson Laboratory. Sox9-CreER[T2] knockin mice[56] and CAG-CAT-eGFP transgenic mice[57] were kindly provided by H. Akiyama (Kyoto University) and J. Miyazaki (Osaka University Medical School), respectively. *Reeler* mutant mice were kindly provided by K. Nakajima (Keio University). Nex-Cre knockin mice were kindly provided by K. A. Nave (Max Planck Institute) and N. Tamamaki (Kumamoto University). *Dab1*[fl/fl] mice were established as described[49]. All mice were maintained and studied according to protocols approved by the Animal Care and Use Committee of The University of Tokyo. For induction of CreER[T2] activity, adult mice were injected intraperitoneally with 0.1–0.3 mg of tamoxifen (Sigma) in corn oil at P60 or pregnant mice were orally administered 5–6 mg of tamoxifen in sunflower oil at E17.

**Antibodies**. Primary antibodies included those to GFP (rabbit, 1:1000 dilution, MBL, 598; chicken, 1:1000, Abcam, ab13970), GFAP (rabbit, 1:1000, Dako, Z0334; mouse, 1:500, Millipore, MAB360), NeuN (mouse, 1:200, Millipore, MAB377), S100β (mouse, 1:500, Sigma, SH-B4; rabbit, 1:1000, Dako, Z0311), S100 (mouse, 1:500, Abcam, ab4066), Sox10 (goat, 1:200, Santa Cruz Biotechnology, sc17342), Cux1 (rabbit, 1:1000, Santa Cruz Biotechnology, sc13024), CTIP2 (rat, 1:2000, Abcam, ab18465), Zbtb20 (mouse, 1:200, Abcam, ab48889), HMGN1 (rabbit, 1:1000, Abcam, ab5212), HMGN2 (rabbit, 1:500, LifeSpan Biosciences, LS-C118756), HMGN3 (rabbit, 1:100, Santa Cruz Biotechnology, sc138955), Id1 (rabbit, 1:200, Biocheck, BCH-1/37-2), and Lef1 (rabbit, 1:1000, Cell Signaling, #22). Secondary antibodies included those conjugated with Alexa Fluor fluorophores to mouse, rabbit, chicken, rat, or goat immunoglobulin G (1:500, Invitrogen).

**Immunostaining**. Mice were anesthetized and intracardially perfused first with phosphate-buffered saline (PBS) and then with 4% paraformaldehyde in PBS. The brain was dissected out, exposed at 4 °C consecutively to 4% paraformaldehyde in PBS for 2 h, 15% sucrose in PBS for 6–12 h, and 30% sucrose in PBS for 24 h, and then embedded and frozen in OCT compound (TissueTEK) and stored at –80 °C. Coronal sections were prepared at a thickness of 12 μm with the use of a cryostat (Microme HM560) and stored at –80 °C. The sections were subsequently exposed to Tris-buffered saline containing 0.1% Triton X-100 (TBST) for 15 min at room temperature, treated with blocking solution (2% donkey serum in TBST) for 1 h at room temperature, and then incubated first with primary antibodies in blocking solution at 4 °C overnight and then for 1 h at room temperature with Alexa Fluor-conjugated secondary antibodies and Hoechst 33342 (1:1000, Molecular Probes) in blocking solution. As for Id1 and Lef1 staining, sections were autoclaved in the Target Retrieval solution (Dako) at 105 °C for 10 min before the first blocking step. The sections were washed in TBST three times for 15 min each time after incubations with antibodies, and they were finally mounted in Mowiol (Calbiochem).

**Quantification of immunofluorescence staining**. We defined an astrocyte nucleus (nucleus of a GFP- or S100-positive cell) as the ROI and quantified the median fluorescence intensity of HMGN1, HMGN2, HMGN3, Zbtb20, Lef1, and Id1 signals in the ROI. The signal intensity was then converted to a scale of 0–100 and plotted.

**CUBIC technique**. The brain was dissected out and exposed to 4% paraformaldehyde at 4 °C for 24 h. Coronal sections (thickness of 400 μm) were prepared with the use of a vibrating microtome (LinearSlicer PRO7, Tedpella), washed in PBS, and then subjected to the CUBIC (clear, unobstructed brain/body imaging cocktails) procedure as described previously[19]. In brief, the sections were incubated at room temperature first with 50% CUBIC1 reagent (25% urea, 25% Quadrol, and 15% Triton X-100) for 6 h and then with 100% CUBIC1 for 6 h. They were then washed three times in PBS for 30–60 min each time before incubation at room temperature first with 50% CUBIC2 reagent (25% urea, 50% sucrose, and 10% triethanolamine in PBS) for 6–12 h and then with Hoechst 33342 (1:1000) in 100% CUBIC2 for 24–48 h.

**2D imaging, 3D imaging, and morphological analyses**. For 2D imaging, fluorescence images were acquired with a laser confocal microscope (Leica TCS-SP5) equipped with a water-immersion objective lens (×63). Confocal images were processed with the Fiji package of ImageJ (NIH). For 2D cell orientation analysis, individual astrocytes in z-projected images were measured for their elongation and orientation angles. These angles were further calculated relative to the brain surface. For 2D Sholl analysis, we used coronal sections with 12 μm thickness and examined branching complexity of astrocytes at a coronal plane which contained their nucleus. Series of confocal images were converted to 8-bit gray-scale images, and the astrocyte processes were manually traced with the use of the Simple Neurite Tracer plugin of Fiji. Traces were z-projected and analyzed with the Sholl Analysis plugin of Fiji.

For 3D imaging, fluorescence images (1.48-μm intervals) were acquired with a laser confocal microscope (Leica TCS-SP5) equipped with an oil-immersion objective lens (×40). Confocal images were processed with Photoshop CS4 software (Adobe), the Fiji package of ImageJ, or FluoRender (University of Utah). For quantification of 3D-cell orientation, individual astrocytes were extracted with the 3D Object Counter plugin of Fiji based on the red (tdTomato) channel of confocal images from Glast-EMTB-GFP;Glast-CreER^T2;Rosa-CAG-LSL-tdTomato mice. The 3D objects of tdTomato-labeled astrocytes were then analyzed with the 3D Suite plugin for measurement of object volume, surface area, compactness, sphericity, convex hull volume, convex hull surface area, solidity, and convexity. The 3D Ellipsoid Fitting plugin was then applied to fit a 3D ellipsoid to objects in a labeled image, and the major axis length, middle axis length, minor axis length, elongation, flatness, and orientation angles relative to the XY, XZ, and YZ planes of the fit ellipsoid were measured with the 3D Suite plugin. The orientation angles relative to the XZ and YZ planes were further calculated relative to the brain surface. After the ROI for the tdTomato object was defined, the microtubule structure within that ROI was extracted with the Image Calculator plugin by subtraction of the GFP signal outside the cell from the original image of the GFP channel. The volume and surface area of these automatically extracted 3D microtubule structures were measured, and they were analyzed directly with the 3D Sholl Analysis plugin of Fiji. We excluded fibrous and perivascular astrocytes from our analysis.

**Morphometric cluster analysis**. We quantified astrocyte morphologies according to the 24 parameters shown in Supplementary Table 1 with the use of the 3D Suite plugin in Fiji. All quantified datasets were analyzed in the statistical environment R version 3.31 (http://www.R-project.org). Prior to morphometric clustering, each dataset was transformed to normalized scores by subtraction of the mean and division by the standard deviation as previously described (21). The MMI was calculated according to the formula: MMI = $[M3^2 + 1]/[M4 + 3(n-1)^2/(n-2)(n-3)]$, where M3 is skewness, M4 is kurtosis, and $n$ is sample size. t-SNE analysis was performed with Rtsne package version 0.11 for R[22] with the values set as follows: seed = 510, theta = 0, max_iter = 100,000, and perplexity = 30. Hierarchical clustering was performed by Ward's method[23], with the optimal number of clusters being determined with Thorndike's procedure[24]. In brief, total within-cluster distance was plotted for the different numbers of clusters, and a sudden marked flattening of the curve determined the optimal number of clusters. Silhouette analysis was applied to estimate the quality of clustering by examination of within-cluster and between-cluster distances[25]. The silhouette width, $s(i)$, of data point $i$ in a given cluster is calculated by the formula: $s(i) = b(i) - a(i)/\max[a(i), b(i)]$, where $a(i)$ is the average cluster distance between $i$ and all other data points in a cluster and $b(i)$ is the smallest average cluster distance between $i$ and all other data points in any other cluster. The value of $s(i)$ is between −1 and 1.

**Electron microscopy**. Electron microscopic analysis was performed as described previously[59] with slight modifications. Female C57BL/6 J, reeler, or Dab1 cKO mice at 7–10 weeks of age were anesthetized with pentobarbital and perfused with saline followed by Karnovski fixative (2% paraformaldehyde (TAAB) and 2.5% glutaraldehyde (Nacalai) in 0.15 M sodium cacodylate buffer containing 2 mM CaCl₂).

The brain was dissected out, exposed at 4 °C overnight to 4% paraformaldehyde in 0.15 M sodium cacodylate buffer containing 2 mM CaCl₂, and coronally sectioned at a thickness of 100 μm with the use of a vibratome (Linear Pro7, Dosaka EM). The slices were washed in ice-cold 0.1 M sodium cacodylate buffer containing 2 mM CaCl₂, incubated for 1 h on ice with the same solution containing 2% OsO₄ and 1.5% potassium ferrocyanide, rinsed with double-distilled water, treated with 1% thiocarbohydrazide for 20 min at room temperature, and stained further with 2% OsO₄ for 30 min at room temperature. They were then stained first with 1% uranyl acetate overnight at 4 °C and then with lead aspartate (0.066 g of lead nitrate in 10 ml of 0.003 M aspartic acid, pH 5.5) at 60 °C for 30 min, dehydrated with a graded series of ethanol solutions (50, 70, 80, 90, 95, 100, and 100%, 5 min each), immersed in propylene oxide twice for 10 min, and mounted with Durcupan resin (Sigma). For SBF-SEM imaging, serial images of astrocytes and synaptic contacts in the primary somatosensory cortex were collected in a layer-specific manner with the use of a scanning electron microscope (Marlin, Zeiss) equipped with a 3View ultramicrotome (Gatan) and at a 50-nm Z-step and an acceleration voltage of 1.4–2.0 kV. Images were captured at a magnification of ×4730 and with 8192 by 8192 pixels for the X and Y axes. For FIB-SEM imaging, serial images of astrocytes and synaptic contacts in the primary somatosensory cortex were collected in a layer-specific manner with the use of a scanning electron microscope (Scios, FEI) equipped with a FIB and at a 20-nm Z-step and an acceleration voltage of 1.5 kV. Images were captured at a magnification of ×17,500 and with 3072 by 2048 pixels for the X and Y axes, respectively. For this imaging condition, the size of a pixel for X, Y, and Z axes is 3.854, 4.891, and 20 nm, respectively. All images were analyzed with RECONSTRUCT[60] and the Fiji package of ImageJ.

**Quantification of synapse ensheathment by astrocytes**. We identified synaptic structures composed of a presynaptic axon containing vesicles (small dots in SEM images), a postsynaptic dendrite containing Golgi bodies (short stripes in SEM images), and a synaptic cleft with a bold black shadow on the dendritic membrane corresponding to the postsynaptic density. Each synapse was examined for the presence or absence of astrocytes at the perimeter of synaptic axon–dendrite interface. Synaptic cleft-apposed structures were inspected through the z-stack and then identified the cell type. Astrocyte was identified by their irregular shapes interdigitating among neuronal profiles, often making contacts with variable portions of the synapse, and by the presence of granules of glycogen distributed over a relatively clear cytoplasm (Fig. 3a). To determine how much of the perimeter of a synaptic axon–dendrite interface was surrounded by astrocytic profiles, we used the procedure described by Ventura et al[61]. On each section of a cross-sectioned synapse, the perimeter had two parts, one at each edge of the synaptic axon–dendrite interface. At the ending caps (the first and last section of the synaptic axon–dendrite interface), the next section was evaluated whether astrocytic processes surrounded the cap. Perimeter of synaptic axon–dendrite interface was approximately calculated by summing 2-fold of section interval (20 nm) over all crossed-sectioned synapse slices, and summing with the length of two caps of synapse. Astrocytic perimeter was measured by summing the section intervals and the length of the caps if there are evident astrocyte contacts (apposed astrocyte area >0.005 μm²) at the edges of synaptic axon–dendrite interface. Then the fraction of edges with astrocytic profiles was determined (Fig. 3b).

**Tissue dissection, dissociation, and FACS**. Young adult Aldh1l1-eGFP mice were killed by cervical dislocation, and layer dissection was performed as described[58]. In brief, the brain was embedded in 4% low-melting point agarose (Lonza), and coronal sections were prepared at a thickness of 250 μm with a vibrating microtome (LinearSlicer PRO7, Tedpella) and collected in ice-cold Dulbecco's modified Eagle's medium (DMEM). Regions of the sections corresponding approximately to cortical layers II/III–IV (upper layers) and to layers V and VI (deep layers) were dissected out under visual guidance and transillumination with a microscope (Zeiss Stemi 2000-C) and were also collected separately into ice-cold DMEM. The separated tissue was dissociated into a single-cell suspension with the use of a neural tissue dissociation kit (Sumilon). Cells were isolated by centrifugation (100×g for 2 min), resuspended in PBS containing 0.1% bovine serum albumin (0.1% BSA/PBS), and passed through filter-top tubes (35 μm, Falcon). Cell sorting was performed with a FACS AriaII or AriaIII cytometer (BD Biosciences), after which the cells were collected in low-binding 1.5-ml tubes containing either 0.1% BSA/PBS for RT-qPCR analysis or RNAlater (Qiagen) for RNA-seq analysis, isolated by centrifugation (600×g for 10 min), and resuspended in 300 μl of TRIzol (Invitrogen) or RNAiso Plus (Takara).

**RNA extraction and RT-qPCR analysis**. Total RNA was isolated from dissected tissue sections or sorted cells with TRIzol (Invitrogen) or RNAiso Plus (Takara) and was then subjected to RT with gDNA Remover and ReverTra Ace (Toyobo). The resulting cDNA was subjected to real-time PCR analysis in a Roche Light-Cycler instrument with Thunderbird SYBR qPCR Mix (Toyobo) or KAPA SYBR fast qPCR Mix (Kapa Biosystems). The abundance of target mRNAs was normalized relative to that of β-actin mRNA. The PCR primers (sense and antisense, respectively) were as follows: Aldh1l1, 5′-AGCTGTGCCCTGAGTAA-3′ and 5′-GTCACAGTCAGCAAAGATGATAAG-3′; Actb (β-actin), 5′-AATAGTCATTC-CAAGTATCCATGAAA-3′ and 5′-GCGACCATCCTCCTCTTAG-3′; Baf53b,

5′-CCAAGGAGCCTGTACGG-3′ and 5′-TATGCCAGGACTTGGAGAC-3′;
*Tubb3* (βIII-tubulin), 5′-ACACAGACGAGACCTACT-3′ and 5′-GCAGACA-
CAAGGTGGTT-3′; *Ctip2*, 5′-CCCTTTCCAGCTCTCTTCC-3′ and 5′-
AGGTCTTTCTCCACCTTGAT-3′; *Cux1*, 5′-CATATCAGCAGAAGCCATACC-
3′ and 5′-ATGGAACCAGTTGATGACG-3′; *Fezf2*, 5′-CTCTACTGA-
CAGCAAACCCA-3′ and 5′-CTTTGCACACAAACGGTCT-3′; *Slc1a3* (Glast), 5′-
CAAGTTCTGCCACCCTAC-3′ and 5′-CACAAATCTGGTGATGCGT-3′; *Rbfox3*
(NeuN), 5′-GCTGATCCTTACCATCACAC-3′ and 5′-CATGGTCCGA-
GAAGGAG-3′; and *Svet1*, 5′-TTTCAGACTATGTTCAAAGCCC-3′ and 5′-
TCATCTATCCTGTTGCTACGAC-3′.

**RNA-seq analysis**. Total RNA from ULA and DLA samples ($n = 3$ brains from 4-
week-old male mice) was extracted as described for RT-qPCR analysis and was
subjected to RT with the use of a SMART-Seq v4 Ultra Low Input RNA Kit for
Sequencing (Clontech). Bar-coded libraries were prepared with a Nextera XT DNA
Library Preparation Kit (Illumina), and single-end 36-bp sequencing was per-
formed with a HiSeq 2500 instrument (Illumina).

**Statistical analysis**. Data are presented as means ± s.d or ± s.e.m. as indicated, and
were analyzed by one-way or two-way analysis of variance (ANOVA) followed by
Bonferroni's post hoc test, by the two-sided Fisher's exact test, with Welch's *t*-test,
or by Student's two-tailed paired *t*-test, as indicated. A $P$ value of <0.05 was
considered statistically significant and the significance is marked by *$P < 0.05$, **$P
< 0.01$, and ***$P < 0.001$. The number of animals in each experiment is stated in the
respective figure legends.

**Data availability**. All relevant data are available upon request. RNA-seq data are
deposited to NCBI GEO database under accession number GSE111916.

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

## Acknowledgements

We thank E. S. Anton, M. Götz, H. Akiyama, J. Miyazaki, and K. Nakajima for mice; T. Horiuchi, K. Imamura, and other members of the Suzuki laboratory for help with sequence analysis; M. Nagao (National Rehabilitation Center for Persons with Disabilities, Japan) and T. Miyata (Nagoya University) for discussion and technical advice; E. A. Susaki and H. R. Ueda (The University of Tokyo) for CUBIC reagents and technical advice; and all members of the Gotoh laboratory for discussion and support. This research was supported by AMED-CREST of the Japan Agency for Medical Research and Development, as well as by AMED and Grants-in-Aid for Scientific Research on Innovative Areas ("Interplay of Developmental Clock and Extracellular Environment in Brain Formation," grant no. JP16H06479), for Scientific Research (S) (grant no. JP15H05773), for a JSPS Research Fellow (grant no. JP11J10035) from the Japan Society for the Promotion of Science (JSPS), and for JSPS KAKENHI grant no. 16H06279. This work was partly supported by the International Research Center for Neurointelligence (WPI-IRCN) at The University of Tokyo Institutes for Advanced Study.

## Author contributions

D.L., B.-J.P., and D.K. designed and performed the experiments, analyzed the data, and wrote the manuscript. T.T. performed cluster analysis. Y.S. conducted RNA-seq analysis. Y.F. performed SEM experiments. S.F. and Y.G. conceived of and coordinated the project, as well as wrote the manuscript. Y.K. provided *Dab1* cKO mice. All authors approved the final version of the manuscript.

## Additional information

**Competing interests:** The authors declare no competing interests.

