## [Peer Review File · Nature Communications]

Reviewers' comments:

Reviewer #1 (Remarks to the Author):

In the manuscript entitled "Layer-specific heterogeneity of astrocytes in the mouse neocortex," Lanjakornsiripan and colleagues describe the morphological and molecular heterogeneity of astrocytes of the mouse somatosensory cortex. Neuronal heterogeneity in the neocortex has been well defined, in terms of morphology, synaptic connections and layer-specific molecular markers. However, whether protoplasmic astrocytes of the mammalian cortex are homogenous or heterogeneous is unknown. Through sparse genetic labeling, the authors demonstrate that astrocytes of the neocortex are not homogeneous in their structure by demonstrating layer-specific morphological differences. Indeed, astrocytes from each cortical layer display distinct sizes and orientations and unique synaptic interaction profiles. Beyond cellular phenotypes, the author also performed transcriptome profiling of astrocytes between different cortical layers to identify transcription factors that are differentially expressed between upper-layer and lower-layer astrocytes. Finally, the authors utilize Reeler mice, in which cortical neuronal layering is perturbed, to show that extrinsic signals may alter layer-specific characteristics of protoplasmic astrocytes of the cortex.

While the authors show convincing evidence for layer-specific morphological differences in astrocytes, there are several major concerns regarding the interpretation of the data and their claims of molecular heterogeneity. These concerns are listed below.

Major Concern:

Astrocyte heterogeneity is a newly emerging concept; thus, it is still somewhat open to interpretation. However, for claims of heterogeneity between protoplasmic cortical astrocytes to be verified one expects to see that either these astrocytes originate from distinct sets of progenitors and/or they perform distinct functions which are non-overlapping. The subtle layer-specific morphological and molecular differences reported here between astrocytes are likely not due to the heterogeneity of astrocytes themselves but due to their ability to reflect the differences in their surroundings. The paper at its current form does not provide sufficient convincing data that can support a claim of astrocyte heterogeneity.

Other Concerns:

- 1) Throughout the manuscript clarify the age and sex of the mice used for each experiment and the rationale for using mice of different ages between different experiments.
- 2) In Figure 3 the authors performed serial block TEM to visualize astrocyte synapse interactions. The quantification of astrocyte-synaptic cleft interactions is not satisfactory. A more complete analysis showing the percent of synapse ensheathment by astrocyte and the preference of pre-versus post-synaptic association are necessary.
- 3) The claim that HMGN expression in astrocytes is different between different cortical layers of the cortex is unconvincing. The images in Figures 5-6 seem to show that HMGN2 is present in all 3 representative images displayed. Representative images of HMGN1, and Zbtb20 at different layers are not shown.
- 4) Similarly, it is hard to make out any differences in the staining of other proposed layer specific astrocytic proteins such as EphA3. As a side note EphA3 antibodies are notoriously unspecific, particularly for staining in tissue. So, it is hard to believe in the reported staining.
- 5) The only gradient visible is diminished GFP expression in the inner cortical layers, which may be a concern since FACS by GFP was used to isolate astrocytes from different layers.
- 6) The authors utilize Reeler mice to determine the effects of perturbations in cortical neuronal layering on astrocyte profiles. These experiments are not well done and not thoroughly analyzed in a way similar to figures 1-2. In particular, authors do not provide control images depicting altered layering of neurons. In bringing the manuscript full circle to test the effect of neuronal layering on astrocytes, is astrocyte morphology and or synaptic interaction different in Reeler^{-/-} mice

compared to controls?

7) Figure S5 is missing and Figure S7 is duplicated as Figure 7.

Reviewer #2 (Remarks to the Author):

In this manuscript, Darin et al. showed that neocortical astrocytes have layer-specific morphology and molecular heterogeneity. They demonstrated that astrocytes in upper-layers and deeper-layers have different morphology and separated them into four subpopulation by unbiased cluster analyses. Using SBF-SEM imaging, they found different synaptic interaction of astrocyte in different layers. By layer-specific RNA-Seq and qPCR, they showed astrocytes in different layer have molecular heterogeneity and this heterogeneity was disturbed in reeler mutant mice. This story provide novel insight into layer specific heterogeneity of neocortical astrocytes.

Major comments:

1. The authors used GLAST promotor to drive tdTomato or microtubule marker expression in a subset of astrocytes. Could GLAST promotor be expressed by different subsets of astrocytes at different cortical layers? The conclusions would be much strengthened if they could use an additional GLAST-independent method to label astrocytes (e.g. dye fill, sparse labeling by virus injection), even if that's done with a small number of samples. Or perhaps the author can analyze images from the literature using different methods.
2. Supplemental table 3, RNAseq data is based on n=1. That would not be suitable for publication at Nature Communications, or perhaps any journal.
3. It has been reported that astrocyte processes favor post-synaptic processes more than pre-synaptic processes. In Figure 3a, it looks like deeper-layer astrocyte processes attach to post-synaptic processes more than upper-layer. Is there differences in astrocyte coverage of pre vs post synaptic compartments across layers?
4. Reeler mutant mice should have an inverted cortex. To test whether astrocyte heterogeneity arise intrinsically or extrinsically induced by the microenvironment, why don't the authors look at the morphological, orientational, or molecular differences they found in reeler mice? They only look at HMGN2 but the images are really difficult for readers to tell what happened.
5. In general the staining images from this manuscript have low signal to noise. It's hard to tell, for example, if the HMGN2 gradient is abolished in reeler mice based on the images.

Minor comments:

1. Page 9 line 173, synaptic cleft in layer 2/3 are "almost entirely" wrapped by astrocytes. The quantification in Figure 3 showed ~0.6 ratio of astrocyte-synaptic cleft interaction though, not "almost entirely". Changing the text to "the majority of synaptic cleft" would be more accurate.

Reviewer #3 (Remarks to the Author):

Astrocytes play a major role in various brain processes and insight into their structural, molecular, and physiological properties is needed to understand how the brain is organized and functions. In this study, Lanjakornsiripan et al. investigate the morphological and gene expression differences among cortical astrocytes populations. Interestingly, they show that astrocytes in discrete cortical lamina have morphological biases and enrichment for certain molecules. Overall, the results of this manuscript help support the concept that astrocytes represent a heterogeneous population of cells whose organization is coordinated with surrounding neuronal populations and circuitry. However, there are major limitations with the current study that significantly dampen enthusiasm for the work. Although the morphological analysis is interesting and largely complete, the corresponding molecular analysis is weak and does not help provide a better understanding of the importance of

diverse lamina-specific astrocyte subpopulations.

Major Points:

1. Abstract/Introduction/Discussion (first paragraph, lines 32-33, 51-53, 264-265). There are several strong statements suggesting that current knowledge in the field indicates that cortical protoplasmic astrocytes represent a 'non-diverse population.' Although historically this is true, current evidence does indicate heterogeneity of cortical astrocytes with respect to their anatomy (layer 1 vs. other layers), molecular properties (transporters, GFAP etc.), and calcium activity (indicated by authors). The authors should revise these statements to be clearer about what is known/not-known and cite related literature. The authors should also reference Tsai et al. Science (2012) which showed region-specific allocation of astrocytes to specific areas of cortex.

2. Overall, the authors performed a thorough characterization of astrocyte morphology in cortex utilizing multiple genetic labeling methods. Cluster analysis using numerous morphological parameters segregated astrocytes into subpopulations that display bias for particular cortical layers. However, several questions arise from the analysis:

a. What part of cortex is being studied for the morphological analysis of astrocytes?

Somatosensory cortex? This needs to be clearly indicated. If not somatosensory cortex, how can the results be reconciled with subsequent molecular analysis?

b. The Sholl analysis appears to indicate that layer II/III astrocytes have larger arbors. However, evidence that that these cells have increased ramification (especially close to the nucleus; i.e. 10-30um range) is not supported. The data suggests similar ramification between these astrocyte subpopulations.

c. The authors need to provide a better justification for why two clustering methods (tSNE and Ward's) are used and reconcile why the two methods identified different numbers of astrocyte subpopulations.

d. Were specific astrocytes selected or excluded from the morphological analysis based on certain criteria (i.e. association with blood vessels, closeness to corpus callosum, etc.)? Inclusion of certain perivascular and white matter astrocytes, which are known to have distinct properties and polarity, could skew analysis and affect the interpretation of the data.

3. In general, the molecular analysis of layer-specific astrocyte subpopulations appears preliminary and is not especially helpful in suggesting functional differences among astrocyte populations in supporting neurons in specific cortical lamina or controlling/organizing specific aspects of cortical circuits. Information also appears to be missing (i.e. Supplement Figure 5 and Supplemental Figure 7 is identical to Figure 6).

a. In general, the target genes selected for qPCR do not provide meaningful insight into specific astrocyte functions. It is unclear why astrocyte genes with known function were not better explored and subjected to further analysis.

b. The RNAseq dataset for ULAs/DLAs needs to be developed and included in the analysis. This information would be useful and provide an unbiased collection of astrocytic mRNAs that are up/down regulated in upper and lower layers.

c. Figure 5. The immunofluorescence results meant to support the qPCR data showing enrichment of EphA3 and Adcy8 in ULA and DLAs, respectively, is not convincing. EphA3 does not look enriched in layer II/III astrocytes (ULAs) and Adcy8 doesn't look enriched in the DLAs shown. In fact, Adcy8 seems most enriched in layer V.

4. The use of Reelin mice for testing how brain microenvironments establish/maintain astrocyte heterogeneity was not particularly convincing considering that loss of Reelin is known to cause major disruptions to the developing cortex and could directly affect the proliferation, migration and maturation of astrocytes. More detailed analysis is needed to verify the utility of the Reelin model for this analysis. Otherwise a more specific model should be used.

Reviewer #1

We thank the reviewer for the careful evaluation of our manuscript and for constructive comments, our responses to which are as follows:

Major Concern:

Astrocyte heterogeneity is a newly emerging concept; thus, it is still somewhat open to interpretation. However, for claims of heterogeneity between protoplasmic cortical astrocytes to be verified one expects to see that either these astrocytes originate from distinct sets of progenitors and/or they perform distinct functions which are non-overlapping. The subtle layer-specific morphological and molecular differences reported here between astrocytes are likely not due to the heterogeneity of astrocytes themselves but due to their ability to reflect the differences in their surroundings. The paper at its current form does not provide sufficient convincing data that can support a claim of astrocyte heterogeneity.

We appreciate this fundamental point raised by the reviewer. However, we believe that statistically significant differences in morphological and molecular properties can provide a basis for distinguishing cellular subpopulations or “heterogeneity,” given that these properties have been commonly applied to the classification of cell types in other lineages and even to astrocytes (John Lin et al. Nat. Neurosci. 2017). In addition, we discovered the difference in the extent of synapse ensheathment between upper-layer and deep-layer astrocytes, which may be related to their functional differences.

Other Concerns:

1) Throughout the manuscript clarify the age and sex of the mice used for each experiment and the rationale for using mice of different ages between different experiments.

We apologize for the insufficient information regarding mouse age and sex in the original manuscript. We have now added these details in the “Methods” section and in the figure legends of the revised manuscript.

2) In Figure 3 the authors performed serial block TEM to visualize astrocyte synapse interactions. The quantification of astrocyte-synaptic cleft interactions is not satisfactory. A more complete analysis showing the percent of synapse ensheathment by astrocyte and the preference of pre- versus post-synaptic association are necessary.

We have now analyzed the percentage of synapse ensheathment by astrocytes and included the results in new Figure 3b. We were able to identify synaptic clefts in the SEM images, but it is difficult to define “presynaptic regions” and “postsynaptic regions” in these images given the ambiguity between synaptic and nonsynaptic regions. We were therefore not able to determine a preference of astrocytes for pre- versus postsynaptic association.

3) The claim that HMGN expression in astrocytes is different between different cortical layers of the cortex is unconvincing. The images in Figures 5-6 seem to show that HMGN2 is present in all 3 representative images displayed. Representative images of HMGN1, and Zbtb20 at different layers are not shown.

In response to this comment, we improved the immunostaining data in new Supplementary Figure 3 by including better representative images of HMGN1, HMGN2, HMGN3, and Zbtb20 at high magnification and resolution.

4) Similarly, it is hard to make out any differences in the staining of other proposed layer specific astrocytic proteins such as EphA3. As a side note EphA3 antibodies are notoriously unspecific, particularly for staining in tissue. So, it is hard to believe in the reported staining.

We attempted to improve the immunostaining results for EphA3 but failed, probably because of the poor quality of the antibodies as pointed by the reviewer. We thus removed these data and instead now include immunostaining for Id1 in new Figure 5, the gene for which showed marked differences in expression in our new RNA-seq results.

5) The only gradient visible is diminished GFP expression in the inner cortical layers, which may be a concern since FACS by GFP was used to isolate astrocytes from different

layers.

We confirmed that astrocytes were specifically isolated with the use of RT-qPCR analysis after isolation of GFP⁺ cells by FACS, as shown in Figure 4.

6) The authors utilize Reeler mice to determine the effects of perturbations in cortical neuronal layering on astrocyte profiles. These experiments are not well done and not thoroughly analyzed in a way similar to figures 1-2. In particular, authors do not provide control images depicting altered layering of neurons. In bringing the manuscript full circle to test the effect of neuronal layering on astrocytes, is astrocyte morphology and or synaptic interaction different in Reeler^{-/-} mice compared to controls?

As suggested by the reviewer, we now include images of neuronal layers in Reeler mice (new Supplementary Figure 4) that show inversion of upper (Cux1⁺) and deep (Ctip2⁺) layer neurons. We also analyzed astrocyte morphology and synaptic interaction in Reeler mice so as to compare them with those in control mice. We indeed found that the layer-specific features of astrocytes, including cell orientation (S100 β staining) and synapse ensheathment (serial SEM), were disturbed in Reeler mice (new Figures 3c and 6). Furthermore, we have now examined the layer-specific properties of astrocytes with the use of *Dab1*^{fl/fl}; *Nex-Cre* mice, in which neuronal layers are disorganized as a result of the lack of Dab1 specifically in postmitotic neurons (new Supplementary Figure 4). We found that gene expression (Id1), astrocyte orientation (S100 β staining), arborization (Glast-EMTB-GFP immunofluorescence) and synapse ensheathment (serial SEM) were also disturbed in these mutant mice as in Reeler mice, providing strong evidence that layer-specific astrocyte heterogeneity is established in a manner dependent on neuronal layers (new Figures 3d, 5, 6, 7).

7) Figure S5 is missing and Figure S7 is duplicated as Figure 7.

We apologize for these mistakes, which have now been corrected in the revised manuscript.

Reviewer #2

We thank the reviewer for the careful evaluation of our manuscript and for constructive comments, our responses to which are as follows:

Major comments:

1. The authors used GLAST promotor to drive tdTomato or microtubule marker expression in a subset of astrocytes. Could GLAST promotor be expressed by different subsets of astrocytes at different cortical layers? The conclusions would be much strengthened if they could use an additional GLAST-independent method to label astrocytes (e.g. dye fill, sparse labeling by virus injection), even if that's done with a small number of samples. Or perhaps the author can analyze images from the literature using different methods.

We agree with the reviewer and have now labeled astrocytes with the use of another reporter mouse line, Sox9-CreER^{T2};CAG-CAT-eGFP and wild-type mice. We obtained essentially the same results, which are included as new Figure 6 and Supplementary Figure 1.

2. Supplemental table 3, RNAseq data is based on n=1. That would not be suitable for publication at Nature Communications, or perhaps any journal.

In response to the reviewer's comment, we have now repeated this experiment three more times. We thus isolated upper- and deep-layer astrocytes, performed RNA-seq analysis, and obtained similar results. We included these new data in Figures 4d, 4e, Tables 1, 2 Supplementary Tables 2, 3, 4, and we believe that they will be a useful resource for future investigations.

3. It has been reported that astrocyte processes favor post-synaptic processes more than pre-synaptic processes. In Figure 3a, it looks like deeper-layer astrocyte processes attach to post-synaptic processes more than upper-layer. Is there differences in astrocyte coverage of pre vs post synaptic compartments across layers?

This is an interesting point, but it is difficult to define "presynaptic regions" and "postsynaptic regions" in the SEM images, given the ambiguity between synaptic and

nonsynaptic regions. We have thus not been able to determine whether astrocytes show a pre- versus postsynaptic preference for association.

4. Reeler mutant mice should have an inverted cortex. To test whether astrocyte heterogeneity arise intrinsically or extrinsically induced by the microenvironment, why don't the authors look at the morphological, orientational, or molecular differences they found in reeler mice? They only look at HMGN2 but the images are really difficult for readers to tell what happened.

As suggested by the reviewer, we analyzed astrocyte morphology and synaptic interaction in Reeler mice and compared them with those in control mice. We indeed found that the layer-specific features of astrocytes, including cell orientation (S100 β staining) and synapse ensheathment (serial SEM), were disturbed in Reeler mice (new Figures 3c and 6). Furthermore, we examined the layer-specific properties of astrocytes with the use of *Dab1^{fl/fl};Nex-Cre* mice, in which neuronal layers are disorganized as a result of the lack of Dab1 specifically in postmitotic neurons (new Figures 3d, 5, 6, 7). We found that astrocyte orientation (S100 β staining), arborization (Glast-EMTB-GFP immunofluorescence), and synapse ensheathment (serial SEM) were also disturbed in *Dab1^{fl/fl};Nex-Cre* mice as in Reeler mice, strongly indicating that layer-specific astrocyte heterogeneity is established in a manner dependent on neuronal layers.

5. In general the staining images from this manuscript have low signal to noise. It's hard to tell, for example, if the HMGN2 gradient is abolished in reeler mice based on the images.

We have now improved the immunostaining data by including better representative images of HMGN1, HMGN2, HMGN3, and Zbtb20 staining at higher magnification and resolution (new Supplementary Figure 3). We also added the immunostaining data of Id1, the gene for which showed marked differences in expression in our new RNA-seq results (Figure 5).

Minor comments:

1. Page 9 line 173, synaptic cleft in layer 2/3 are “almost entirely” wrapped by astrocytes.

The quantification in Figure 3 showed ~0.6 ratio of astrocyte-synaptic cleft interaction though, not “almost entirely”. Changing the text to “the majority of synaptic cleft” would be more accurate.

We changed the text accordingly. We also replaced this figure with one showing the extent of synaptic ensheathment in the revised manuscript.

Reviewer #3

We thank the reviewer for the careful evaluation of our manuscript and for constructive comments, our responses to which are as follows:

Major Points:

1. Abstract/Introduction/Discussion (first paragraph, lines 32-33, 51-53, 264-265). There are several strong statements suggesting that current knowledge in the field indicates that cortical protoplasmic astrocytes represent a ‘non-diverse population.’ Although historically this is true, current evidence does indicate heterogeneity of cortical astrocytes with respect to their anatomy (layer 1 vs. other layers), molecular properties (transporters, GFAP etc.), and calcium activity (indicated by authors). The authors should revise these statements to be clearer about what is known/not-known and cite related literature. The authors should also reference Tsai et al. Science (2012) which showed region-specific allocation of astrocytes to specific areas of cortex.

We apologize for the inadequate description of current knowledge regarding the heterogeneity of cortical protoplasmic astrocytes (page 3, lines 1 - 11). We have now rewritten the Introduction so as to refer appropriately to the relevant literature. We also now cite the study of Tsai et al. (2012) (page 3, line 6).

2. Overall, the authors performed a thorough characterization of astrocyte morphology in cortex utilizing multiple genetic labeling methods. Cluster analysis using numerous morphological parameters segregated astrocytes into subpopulations that display bias for particular cortical layers. However, several questions arise from the analysis:

- a. What part of cortex is being studied for the morphological analysis of astrocytes?

Somatosensory cortex? This needs to be clearly indicated. If not somatosensory cortex, how can the results be reconciled with subsequent molecular analysis?

The morphological analysis was indeed performed with the primary somatosensory cortex. Although this was mentioned in the text, it was not stated in the Methods section of the original manuscript, an omission that has now been corrected in the revised manuscript.

b. The Sholl analysis appears to indicate that layer II/III astrocytes have larger arbors. However, evidence that these cells have increased ramification (especially close to the nucleus; i.e. 10-30um range) is not supported. The data suggests similar ramification between these astrocyte subpopulations.

The reviewer is correct. The greater number of crossing points in the Sholl analysis cannot be ascribed to increased ramification. We have thus now avoided the term “ramification” and simply describe the difference in the density of large arbors of >30 μm (page 5, lines 4 - 7).

c. The authors need to provide a better justification for why two clustering methods (tSNE and Ward’s) are used and reconcile why the two methods identified different numbers of astrocyte subpopulations.

Ward’s clustering analysis is more commonly used for identifying distinct subpopulations than is t-SNE, which is not a clustering method but is commonly used for “visualization” of high-dimensional data in a space of reduced dimensionality (2D, in this case) in such a way that similar objects are modeled by nearby points and dissimilar objects by distant points. We have now avoided the term “clustering” in describing t-SNE analysis in the revised manuscript (page 5, line 21 – 25).

d. Were specific astrocytes selected or excluded from the morphological analysis based on certain criteria (i.e. association with blood vessels, closeness to corpus callosum, etc.)? Inclusion of certain perivascular and white matter astrocytes, which are known to have distinct properties and polarity, could skew analysis and affect the interpretation of the

data.

We excluded fibrous and perivascular astrocytes, which we believe are morphologically very distinct from protoplasmic astrocytes. We now mention this point in the method section of the revised manuscript (page 13, line 27).

3. In general, the molecular analysis of layer-specific astrocyte subpopulations appears preliminary and is not especially helpful in suggesting functional differences among astrocyte populations in supporting neurons in specific cortical lamina or controlling/organizing specific aspects of cortical circuits. Information also appears to be missing (i.e. Supplement Figure 5 and Supplemental Figure 7 is identical to Figure 6).

According to the suggestion, we improved the molecular analysis part of the study by performing triplicate RNA-seq experiments and examined the layer-specific distribution of more candidate genes. We thus newly identified that Id family members are highly expressed in layer V/VI astrocytes than in layer II/III astrocytes (so as Sparc, an astrocyte-derived inhibitor of excitatory synapse formation) whereas Mertk, which mediates synapse elimination, is highly expressed in layer II/III astrocytes than in layer V/VI astrocytes (new Figures 4d, 4e, 5, Tables 1, 2, Supplementary Tables 2, 3, 4). We believe that these new data strengthen the conclusion of this study.

We also apologize for the mistake in figure labeling in the original manuscript.

a. In general, the target genes selected for qPCR do not provide meaningful insight into specific astrocyte functions. It is unclear why astrocyte genes with known function were not better explored and subjected to further analysis.

We have now included information regarding molecules such as Sparc, Id, Connexin43, Fgfr3, Fmo1, Dio2 and Mertk, that may perform specific functions in astrocytes (page 7, line 23 – 30, Tables 1, 2).

b. The RNAseq dataset for ULAs/DLAs needs to be developed and included in the analysis. This information would be useful and provide an unbiased collection of astrocytic mRNAs that are up/down regulated in upper and lower layers.

We have now repeated the ULA/DLA isolation and RNA-seq experiments three more times and included these data in the revised manuscript. As suggested by the reviewer, we believe that the results will be a valuable resource for future investigations (new Figures 4d, 4e, 5, Tables 1, 2, Supplementary Tables 2, 3, 4).

c. Figure 5. The immunofluorescence results meant to support the qPCR data showing enrichment of EphA3 and Adcy8 in ULA and DLAs, respectively, is not convincing. EphA3 does not look enriched in layer II/III astrocytes (ULAs) and Adcy8 doesn't look enriched in the DLAs shown. In fact, Adcy8 seems most enriched in layer V.

We attempted to improve the immunostaining results for EphA3 and Adcy8 but failed, probably because of the poor quality of the antibodies. We thus removed these data and instead now include immunostaining for Id1, the gene for which showed marked differences in expression in our new RNA-seq results (new Figure 5).

4. The use of Reelin mice for testing how brain microenvironments establish/maintain astrocyte heterogeneity was not particularly convincing considering that loss of Reelin is known to cause major disruptions to the developing cortex and could directly affect the proliferation, migration and maturation of astrocytes. More detailed analysis is needed to verify the utility of the Reelin model for this analysis. Otherwise a more specific model should be used.

We agree with the reviewer regarding the possible roles of Reelin in regulation of the proliferation, migration, and maturation of astrocytes. We therefore investigated *Dab1^{fl/fl};Nex-Cre* mice, in which *Dab1* is specifically deleted in postmitotic neurons, resulting in disorganization of cortical neuronal layers. We indeed found that layer-specific properties of astrocytes such as morphology, arborization and synapse ensheathment were disrupted in *Dab1^{fl/fl};Nex-Cre* mice as they were in Reeler mice (new Figures 3d, 5, 6, 7). These results provide strong support for a role of neurons in establishment of layer-specific astrocyte heterogeneity.

REVIEWERS' COMMENTS:

Reviewer #1 (Remarks to the Author):

The authors provide some new data and analyses based on reviewers' comments. However, my major concern still remains. "Heterogeneity" as characterized by subtle albeit statistically significant differences is false advertising. There are morphological and molecular differences which seem to be dictated by the surrounding neuronal environment. The astrocytes may in fact be the same "subtype" and just have slightly different characteristics influenced by neurons.

Other concerns:

1- It is unclear how the data was generated in Figure 3b for "quantification of astrocyte ensheathment of synapse". What does the number 1.00 represent? Is that the percentage of an individual synapse that is contacted by an astrocyte? Pre and post-synaptic regions are easily defined in the images presented by the authors, and this quantification should also be included. For example, it seems quite apparent in the images that Layer VI astrocytes prefer to contact the postsynaptic side, whereas Layer II/III contact both. This is a potentially interesting finding that is overlooked. Also, there are no representative images for the other conditions (Reeler, etc.).

2- In general all analyses, in particular the the orientation analysis is not as well done here as it is in Figure 1 for the Reeler mice. They use S100b staining for this analysis, rather than a GFP fill of the astrocyte as in the earlier figures. No images of synapse ensheathment are provided.

3- They don't examine astrocyte territory volume. Throughout the paper, it is not clear what thickness of sections they use for immunostaining (only mention for CUBIC technique where they use 400um). To make claims about changes in astrocyte branching complexity, they must ensure that the entire astrocyte is imaged.

Reviewer #2 (Remarks to the Author):

The authors have addressed my concerns and I would like to recommend publication of this manuscript.

Reviewer #3 (Remarks to the Author):

This revised manuscript is significantly improved. However, I have some remaining comments.

1. Page 4, to better reflect the quantitative results in Figure 1d, it is suggest to reword the following sentence, "Quantification of 3D cell orientation also revealed that the average orientation angle relative to the pial surface of layer II/III astrocytes was near 90 degrees whereas that of layer VI astrocytes was near 0 degrees," to "Quantification of 3D cell orientation also revealed that the average orientation angle relative to the pial surface of layer II/III astrocytes was closer to 90 degrees whereas that of layer VI astrocytes was closer to 0 degrees."

2. Figure 3a-b, add units to Y-axis label.

3. Main text (around lines 180-186) and referring to Figures 4a-b. Need to mention the reasoning for using CTIP2 and Cux markers for immunolabeling and Cux, Svet1, Fezf2, and CTIP2 markers for RT-qPCR analysis.

4. To better confirm the utility of the RNAseq analysis, protein validation of a ULA-enriched target gene should be performed (i.e. immunolabeling of Chrd11).

5. Line 244 of the discussion. To be clearer about what was found in the study, the authors should add more detail to the phrase "synaptic interaction" to read "structural interactions with synapses" as no functional data is being provided.

Point-by-point response to the reviewers' comments

Reviewer #1

The authors provide some new data and analyses based on reviewers' comments. However, my major concern still remains. "Heterogeneity" as characterized by subtle albeit statistically significant differences is false advertising. There are morphological and molecular differences which seem to be dictated by the surrounding neuronal environment. The astrocytes may in fact be the same "subtype" and just have slightly different characteristics influenced by neurons.

In order to address the reviewer's concern regarding the use of the term "heterogeneity" to describe our findings, we have removed the term "heterogeneity" from the title and the results and have instead used the phrase "morphological and molecular differences". According to this correction, we have changed the title of the manuscript from "Layer-specific heterogeneity of astrocytes in the mouse neocortex and its dependence on neuronal layers" to "Layer-specific morphological and molecular differences in neocortical astrocytes and their dependence on neuronal layers".

Other concerns:

1- It is unclear how the data was generated in Figure 3b for "quantification of astrocyte ensheathment of synapse". What does the number 1.00 represent? Is that the percentage of an individual synapse that is contacted by an astrocyte? Pre and post-synaptic regions are easily defined in the images presented by the authors, and this quantification should also be included. For example, it seems quite apparent in the images that Layer VI astrocytes prefer to contact the postsynaptic side, whereas Layer II/III contact both. This is a potentially interesting finding that is overlooked. Also, there are no representative images for the other conditions (Reeler, etc.).

We have now added this information to the text (p. 22 line 23 – p. 23 line 9).

We agree with the reviewer that pre-post preference is a potentially interesting finding, and we had tried to address it but failed. Pre- and post-synaptic regions were easily defined within the immediate vicinity of synapse as pointed by the reviewer. However,

we found that a single neuron-astrocyte contact site sometimes spreads widely from synapse toward axon/dendrite, which makes the precise quantification of pre-post preference difficult, given the lack of absolute criteria to define the ending of pre/post synaptic regions. We have thus decided to include only the quantification of synaptic ensheathment, which does not depend on the issue mentioned above at least in our dataset. Regarding representative images, we do not feel appropriate to pick “representative images” among reeler and Dab1 cKO samples given their great variation.

2- In general all analyses, in particular the the orientation analysis is not as well done here as it is in Figure 1 for the Reeler mice. They use S100b staining for this analysis, rather than a GFP fill of the astrocyte as in the earlier figures. No images of synapse ensheathment are provided.

We used S100 β staining for the orientation analysis of astrocytes in reeler and Dab1 cKO mice, given that the S100 β staining method is relevant and comparable with the “GFP fill” method (Figure 1d and 7c). Also, the GFP fill method is not applicable to Dab1 cKO mice, given that both GFP fill mice (Glast-CreER^{T2};Rosa-CAG-LSL-tdTomato mice) and Dab1 cKO mice (Dab1^{fl/fl};Nex-Cre) use Cre recombinase. Therefore, we instead performed S100 β staining for astrocyte orientation analysis in Dab1 cKO mice. We believe that the claim of the manuscript has been supported rationally by our current dataset.

3-They don't examine astrocyte territory volume. Throughout the paper, it is not clear what thickness of sections they use for immunostaining (only mention for CUBIC technique where they use 400um). To make claims about changes in astrocyte branching complexity, they must ensure that the entire astrocyte is imaged.

We did examine astrocyte territory volume in some experiments (please see Figure 1c) but not others. The information about the thickness of sections has now been included in the manuscript (p. 21 line 7). In Figure 8, we used coronal sections with 12 μ m thickness and examined branching complexity of astrocytes at a coronal plane which

contained their nucleus. They did not cover the entire processes, but the data (Figure 8c, control) is consistent with that in Figure 1e where entire astrocytes were imaged. We have now explicitly mentioned such experimental conditions in the main text (p. 13 lines 12 – 14).

Reviewer #2

The authors have addressed my concerns and I would like to recommend publication of this manuscript.

We greatly appreciate the positive comment on our manuscript by the reviewer.

Reviewer #3

This revised manuscript is significantly improved. However, I have some remaining comments.

We thank the reviewer for the positive and constructive comments, our responses to which are as follows.

1. Page 4, to better reflect the quantitative results in Figure 1d, it is suggest to reword the following sentence, "Quantification of 3D cell orientation also revealed that the average orientation angle relative to the pial surface of layer II/III astrocytes was near 90 degrees whereas that of layer VI astrocytes was near 0 degrees," to "Quantification of 3D cell orientation also revealed that the average orientation angle relative to the pial surface of layer II/III astrocytes was closer to 90 degrees whereas that of layer VI astrocytes was closer to 0 degrees."

We agree and have now reworded the sentence following the reviewer's comment (p. 4 lines 29 – 30).

2. Figure 3a-b, add units to Y-axis label.

We apologize for these mistakes, which have now been corrected in the revised figure 3c – e.

3. Main text (around lines 180-186) and referring to Figures 4a-b. Need to mention the reasoning for using CTIP2 and Cux markers for immunolabeling and Cux, Svet1, Fezf2, and CTIP2 markers for RT-qPCR analysis.

Following the reviewer's comment, we have mentioned the markers which we have used in the analysis as follows: "Immunostaining as well as reverse transcription (RT) and quantitative polymerase chain reaction (qPCR) analysis of upper-layer and deep-layer neuronal markers (Cux1 and Ctip2 in both analyses, and Svet1 and Fezf2 in RT-qPCR analysis for further confirmation) verified the effectiveness of the sample preparation procedure (Fig. 4a, b)." (p. 7, lines 17 – 19).

4. To better confirm the utility of the RNAseq analysis, protein validation of a ULA-enriched target gene should be performed (i.e. immunolabeling of Chrd11).

This is a great advice. As suggested by the reviewer, we performed an immunohistochemical analysis for the ULA-enriched gene Lef1. We found that the intensity of Lef1 immunostaining was indeed higher in ULAs (Layer II/III) than in DLAs (Layer VI). Moreover, this layer-specific expression of Lef1 protein was perturbed in Dab1 mutant mice. These new data have now been included in the manuscript as Figure 5a-f. We have also mentioned the data in the text (p. 8, lines 4 – 7; p. 9, lines 6 – 7). We thank the reviewer for the constructive comment.

5. Line 244 of the discussion. To be clearer about what was found in the study, the authors should add more detail to the phrase "synaptic interaction" to read "structural interactions with synapses" as no functional data is being provided.

As suggested by the reviewer, we have now rephrased “synaptic interaction” to “structural interactions with synapses” (p. 6, line 21; p. 9, line 20; p. 10, line 10; p. 18, line 2).

We would like to thank the reviewers again for their very constructive and helpful suggestions.